# SCALABLE ROBUST FEDERATED LEARNING WITH PROVABLE SECURITY GUARANTEES

## ABSTRACT

Federated averaging, the most popular aggregation approach in federated learning, is known to be vulnerable to failures and adversarial updates from clients that wish to disrupt training. While median aggregation remains one of the most popular alternatives to improve training robustness, relying on secure multi-party computation (MPC) to naïvely compute the median is unscalable. To this end, we propose a secure and efficient approximate median aggregation with MPC privacy guarantees in the multi-silo setting, e.g., across hospitals, with two semi-honest non-colluding servers. The proposed method protects the confidentiality of client gradient updates against both semi-honest clients and servers. Asymptotically, the cost of our approach scales only linearly with the number of clients, whereas the naïve MPC median scales quadratically. Moreover, we prove that the convergence of the proposed federated learning method is robust to a wide range of failures and attacks. Empirically, we show that our method displays similar robustness properties as the median while converging faster than the naïve MPC median for even a small number of clients.

## 1 INTRODUCTION

Federated Learning (FL) (McMahan et al., 2017) has emerged as a leading approach for training shared models among clients who do not wish to release their datasets publicly. While FL avoids releasing client data, revealing client models in the clear creates privacy vulnerabilities, e.g., using model inversion attacks (Fredrikson et al., 2015). To this end, global model aggregation is often implemented using Secure Multi-Party Computation (MPC) to improve privacy (Bonawitz et al., 2016). MPC (Yao, 1986; Goldreich et al., 1987) is a cryptographic primitive that makes it possible to evaluate a function on encrypted inputs while ensuring that the only information revealed throughout the computation is the final output of the function. Thus, to avoid revealing each client's local weights, clients encrypt their weights before sending them to the central server. MPC allows the central server to recover a single aggregate update that it sends back to individual clients. Specialized MPC protocols, that allow the server to recover the average of client updates have been widely deployed for Federated Learning.

In addition to privacy, robustness is a major concern in federated learning implementations (Kairouz et al., 2019). The distributed nature of the computation creates a variety of vulnerabilities to adversarial training attacks and client failures such as hardware, software, data, and communication errors (Xie et al., 2019a; 2021). One way to improve robustness is to replace an aggregation procedure that computes means with a more robust aggregate such as the median (Yin et al., 2018). Using the median, up to half of inputs can be from a faulty distribution before the median fails as a robust average estimator. While median aggregation alone is insufficient to guarantee Byzantine robustness (Xie et al., 2020) – median-based robustness guarantees are sufficient against most published attacks and failures. Unfortunately, the best known approaches for median aggregation with MPC are slow, requiring computation that grows quadratically with the number of clients (Tueno et al., 2019).

Our work is motivated by applications to multi-silo federated learning (Kairouz et al., 2019), designed for learning across institutions. For example, an individual hospital may be limited to using only their own patients' data, and sharing patient information among hospitals is generally difficult due to privacy and intellectual property concerns. Here, federated learning can provide a mechanism for widely distributed medical institutions to jointly train a model.

The main contribution of this work is a **new approximate-median aggregation** for robust, privacy-preserving federated learning in the multi-silo, two-server setting. Our approach is the first to have all of the following features:

- **Robustness against failures.** By relying on (an approximation of) the median of client updates, we achieve robust estimates even in settings where updates sent by up to half the clients are faulty. In addition, we empirically evaluate robustness against noise distributions that take the form of bit flips, label switching, and Gaussian noise in up to half of the clients.
- **Preserving Client Privacy via Secure Computation.** We prove that any subset of clients colluding with each other and with one of the two servers obtains no information about the private inputs of other clients, beyond the aggregated (approximate) median that is returned at each step. This provides meaningful privacy to every client and is a meaningful defense, e.g., against model-inversion attacks that take advantage of individual client updates.
- **Scalability.** The running time of our protocol increases linearly with the number of clients; matching the order-wise scaling (with number of clients) of the standard averaging aggregation. This is in contrast to a quadratic factor blowup for a federated learning system that implements median naïvely in MPC. We empirically evaluate the convergence of our system on various datasets, models and under different types of faults.

## 2    Related Work

Federated Learning (McMahan et al., 2017) is a widely adopted technique for training models with data from edge devices. In its plainest form, client updates are sent without encryption to the central server at each round, and the global model is sent back to each client. However, this basic implementation of a distributed machine learning system is not robust against privacy attacks or failures. For instance, (Wang et al., 2019; Geiping et al., 2020) show how model inversion attacks can be successful in a federated learning setting. (Bonawitz et al., 2017; 2016) provide a protocol for secure aggregation of the client updates and computation of the global model in both a semi-honest and malicious setting, with increased runtime and communication overhead. In addition to encryption of client updates, existing work has also considered robustness to failures and Byzantine client attacks, e.g., (Xie et al., 2019a; 2021) prune faulty client updates by assigning each update a score, and only aggregating the updates that have the top scores. Other methods of providing byzantine robustness including trimmed mean (Xie et al., 2019b), Krum (He et al., 2020; So et al., 2021), and norm-based elimination (Gupta et al., 2021) have been studied. In this paper, we strengthen the restriction on client faults and only consider semi-honest clients. Although a semi-honest setting may seem restrictive by forcing parties to follow the dictated protocol, it applies to settings where parties are assumed to have no ill intentions. Furthermore, developing semi-honest protocols are a step towards developing protocols in the malicious security model, where more expensive cryptographic primitives such as zero-knowledge proofs are added on top of the existing semi-honest protocol.

Secure Multi-Party Computation (MPC) provides a way for multiple parties to compute a function on their individual data and only reveal the final output of the function. (Bonawitz et al., 2016; 2017) provide a method for federated learning where client updates have a pairwise additive mask, such that when all the encrypted client updates are summed together, the masks cancel out. In (Tueno et al., 2019), the authors propose a rank computation system where clients send a pairwise masked version of their data point to the central server. The central server uses pairwise comparisons in MPC to calculate the desired ranked element in the clients' combined data. However, pairwise comparisons are expensive in MPC, as additional values for each unique comparison must be generated in advance. Furthermore, agreeing upon pairwise additive masks grows quadratically with the number of parties.

The generalization of a central server to two non-colluding central servers is not a new idea in privacy-preserving analytics and machine learning. For instance, (Corrigan-Gibbs & Boneh, 2017) introduce a cryptographic protocol called secret-shared non-interactive proofs, and show that these can be used to construct an efficient privacy preserving system for the release of aggregate statistics with guarantees on security in the malicious setting. The setting of the paper consists of a small number of servers jointly computing aggregate statistics over private client data. (Mohassel & Zhang, 2017) introduce efficient distributed ML protocols that utilize a two server abstraction and MPC versions of ML computations. However they do not seek to provide robustness. To our knowledge, (He et al., 2020) is the first work to propose an FL system that combines both robustness and privacy, using private calculation of the distance-based aggregation. They also make the assumption

of the central server being two non-colluding central servers, and utilize client side secret sharing of models and server side two party MPC. The two central server setting allows their protocol to scale well with number of parties, compared to a single central server setting. However, their protocol *leaks* the pairwise distance of client updates to one of the servers, which may be undesirable.

Our proposed approach uses the 2-Party MPC comparison protocols presented in (Rathee et al., 2020). We also adopt a pairwise comparison based computation of the $k$-th ranked element proposed in (Tueno et al., 2019) and a bucketing approximate median idea proposed in (Corrigan-Gibbs & Boneh, 2017), and apply them to the setting of distributed machine learning. Finally, we note that this work focuses on security and robustness of centralized model training. Existing work has shown that personalization and mitigating data heterogeneity can be addressed by fine-tuning the robust centralized model (Li et al., 2021). These details are left for future work.

## 3 Preliminaries

**Federated Learning.** We denote $\mathbb{N}$ as the set of positive integers and $\mathbb{R}_+$ as the set of non-negative real numbers. Given $n \in \mathbb{N}$, we denote $[n] = \{1, \ldots, n\}$. We use bold lowercase letters, e.g., $\boldsymbol{w} \in \mathbb{R}^d$, to denote vectors, and $\boldsymbol{w}(k)$ to denote its value at the $k^{th}$ dimension. We consider a two-server multi-client FL system, let $S_0, S_1$ denote the two central severs, and $c_i$ for $i \in [n]$ represent the clients, where $n$ is the total number of clients. Client $c_i$ holds dataset $D_i$. The agreed upon model has $d$ parameters. $\boldsymbol{w}_g$ denotes the global model sent from servers to clients, while $\boldsymbol{w}_i$ denotes the local model held by client $i$. All parties agree beforehand to run $r$ cycles of FL. The loss function for client $c_i$ is $\hat{F}_i : \mathbb{R}^d \to \mathbb{R}_+$. Usually, because clients have different data, the loss functions $\hat{F}_i$ are different. Therefore, the goal of our setting is to minimize the averaged loss across all clients, i.e., $\arg\min_{\boldsymbol{w}} \frac{1}{n} \sum_{i=1}^{n} \hat{F}_i(\boldsymbol{w})$.

We first detail a basic federated learning system with one central server $S$ and $n$ clients. Clients are generally considered to be edge nodes and do not communicate with each other. In one iteration, the system follows the following generic protocol:

1. For $i \in [n]$, client $c_i$ receives global model $\boldsymbol{w}_g$ from central server $S$.
2. For $i \in [n]$, client $c_i$ performs step(s) of Stochastic Gradient Descent of $\boldsymbol{w}_g$ on its dataset $D_i$ to obtain new local model $\boldsymbol{w}_i$. They then send this model to $S$.
3. Upon receiving all client updates, $S$ aggregates them to obtain the new global model. The new global model is computed as $\boldsymbol{w}'_g = \frac{1}{n} \sum_{i=1}^{n} \boldsymbol{w}_i$.
4. $S$ sends $\boldsymbol{w}'_g$ to all the clients.

**Secure Multi-Party Computation.** Secure Multi-party Computation (MPC) (Goldreich et al., 1987) is a cryptographic primitive which allows $n$ parties, each with input value $x_i$, to jointly compute a mutually agreed upon function $f(x_1, x_2 \ldots x_n)$, with the guarantee that no party learns anything other than what can be inferred from its own input and the joint output.

One of the most prominent techniques used in MPC is secret sharing. A $(t, n)$ secret sharing involves splitting a secret $s$ into $n$ shares, then distributing a share to each party, such that any subset of $t - 1$ parties cannot learn anything about $s$, while any $t$ or more parties can jointly reconstruct $s$. In this paper, we make use of $(2, 2)$ additive secret sharing over power-of-two sized integer rings, which involves splitting $s \in \mathbb{Z}_{2^\ell}$ into $s_1, s_2 \in \mathbb{Z}_{2^\ell}$ s.t. $s_1 + s_2 = s$, and individually $s_i, i \in \{0, 1\}$ has a uniform random distribution. We use $\langle s \rangle$ to denote secret sharing of $s$ over $\mathbb{Z}_{2^\ell}$, and $\langle s \rangle_0, \langle s \rangle_1$ to denote the shares of the two parties. We use $\mathcal{F}^\ell_{\text{Comp}}, \mathcal{F}^\ell_{\text{Equal}}$ to refer to the "greater than" and "equals to" ideal functionalities respectively, i.e. where parties have access to trusted functionalities computing the said functions and use the same to describe our protocols. To implement the protocols for $\mathcal{F}^\ell_{\text{Comp}}, \mathcal{F}^\ell_{\text{Equal}}$, we make use of protocols proposed in CrypTFlow2 (Rathee et al., 2020). CrypTFlow2 provides a customized 2-party protocol for solving the millionaire's, and hence also the comparison/equality problems in $\mathbb{Z}_{2^\ell}$ (more details on all this in Appendix C).

## 4 Problem Description

In this section, we introduce a baseline solution to secure and robust FL. Median provides well-studied robustness against outliers compared to mean (Xie et al., 2018). In order to make our MPC protocols for this efficient, we abstract the central server in FL to two non-colluding central servers. The servers each receive one of the $(2, 2)$ secret shares of the client updates, and perform 2-party based secure median computation. Figure 1 provides a representation of our model.

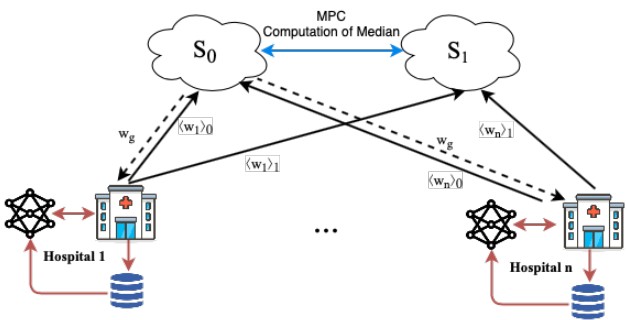

Figure 1: Our proposed two server FL system. Hospitals compute local models $w_i$, and secret share them among the two servers. Servers securely compute the dimension-wise median and end up with the global model. Each hospital receives the global model from $S_0$, and repeats the cycle.

### 4.1 Security Threat Model

We model the adversary using the semi-honest/honest-but-curious model, whereby it is assumed that the parties controlled by the adversary will follow the protocol specifications, but try to learn more information about other parties' data using the protocol transcript. We also assume static corruption, i.e., that the adversary chooses the parties to corrupt at the beginning of the protocol and cannot change them once protocol execution starts. In addition, we assume the two servers $S_0, S_1$ do not collude with each other. The adversary is then allowed to corrupt any arbitrary subset of the clients and at most one of the servers - and once corrupted gets to see their internal state as well as the transcript of their communication. Furthermore, we are concerned with the multi-silo setting, where clients are assumed to be large institutions and thus client drop out or limitations on computational resources are not considered.

### 4.2 Baseline solution

We propose a baseline solution for computing the exact median in our two-server multi-client setting. In order to send their updates to the servers, clients additively secret-share their updates and sends one share to each server. Once all the clients have done so, the two servers now have a $(2, 2)$ additive secret share of all the clients' updates and perform a MPC based median computation. We take inspiration from the ideas in Tueno et al. (2019) to design a 2-party secure protocol for median computation by breaking it up into pairwise comparisons between all elements and some number of equality checks. In Appendix A, we provide a formal description of our algorithm in the $\mathcal{F}^\ell_{\text{Comp}}, \mathcal{F}^\ell_{\text{Equal}}$ hybrid (described in Algorithm 2), along with a correctness sketch and other details for the same. Our final protocol is then obtained by replacing $\mathcal{F}^\ell_{\text{Comp}}, \mathcal{F}^\ell_{\text{Equal}}$ with CrypTFlow2 (Rathee et al., 2020) based secure comparison/equality protocols (described in detail in Appendix C). We refer to this protocol as "pairwise comparison based median" in the rest of the paper.

**Complexity.** The complexity of this protocol is at least $dn(n-1)/2$ calls to $\mathcal{F}^\ell_{\text{Comp}}$, in addition to $dn$ calls to $\mathcal{F}^\ell_{\text{Equal}}$. Note that the number of secure comparisons/equalities grows quadratically with the number of clients $n$. In the next section, we will introduce our approximate median protocol, whose number of secure operations is independent of $n$. We provide a brief summary in Appendix C on why we use number of secure comparisons as a metric for complexity.

## 5 Bucketing based median computation

In this section, we propose an approximate median, which provides increased computational efficiency, to serve as the replacement of the exact median computation. We provide an abridged version of the protocol in Algorithm 1 and a formal description of the protocol in Algorithm 3.

**Summary of the key idea.** Divide the set of possible values that client updates may take into a fixed set of buckets. Let $b$ designate the number of buckets. Next, instead of sending actual updates to the server, each client will send a unit vector that has a 1 in the bucket containing the client's update, and 0 everywhere else. Upon receiving all client updates, the (two) servers build a frequency distribution of client updates and use this to find the bucket which contains the median value. The median is then

---

**Algorithm 1** Succinct version of our bucketed FL system

Input: For $i \in [n]$, $c_i$ holds dataset $D_i$.
$d =$ model size, $b =$ number of buckets, $p$ are public parameters.

---

$\forall i \in [n]$, client $c_i$ does:

1. Receive global model $\boldsymbol{w}_g$ and bucket ranges $B$ from server $S_0$.

2. Compute local model $\boldsymbol{w}_i$ as $SGD(\boldsymbol{w}_g, D_i)$.

3. Compute $Bucketize(w_g, B, b, \boldsymbol{w}_i)$ and obtain bucketized local model $e_i$.

4. Creates $(2, 2)$ shares $\langle \boldsymbol{e}_i \rangle_0$, $\langle \boldsymbol{e}_i \rangle_1$ of $\boldsymbol{e}_i$.

5. Send $\langle \boldsymbol{e}_i \rangle_0$ to $S_0$, send $\langle \boldsymbol{e}_i \rangle_1$ to $S_1$.

$\forall j \in \{0, 1\}$, server $S_j$ does:

1. Receive share $\langle \boldsymbol{e}_i \rangle_j$ from client $c_i, \forall i \in [n]$.

2. Add shares $\langle \boldsymbol{e}_1 \rangle_j \ldots \langle \boldsymbol{e}_n \rangle_j$ to obtain $\langle \sum_{i=1}^n \boldsymbol{e}_i \rangle_j$.

3. Call 2-party secure protocol $\mathcal{F}^\ell_{\text{Comp}}(\langle \sum_{i=1}^n \boldsymbol{e}_i \rangle_j, \lceil \frac{n}{2} \rceil)$ and obtain $d$-length vector $\boldsymbol{m}$, where $\boldsymbol{m}(k)$ indicates which bucket holds the median for dimension $k \in [d]$.

4. $S_0$ calls $Quantize(\boldsymbol{w}_g, B, b, \boldsymbol{m})$ and obtains new global model $\boldsymbol{w}'_g$.

5. $S_0$ computes new bucket range $B'$ as $2\|(\boldsymbol{w}'_g - \boldsymbol{w}_g)\|_1 + p$.

6. $S_0$ releases $\boldsymbol{w}'_g$ and $B'$ to all clients.

---

approximated by the middle value of the range of this bucket. All this while, servers only obtain $(2, 2)$ additive secret shares of client vectors, and rely on MPC to perform the computation outlined above and find the median. We now describe these steps in additional detail.

We "bucketize" clients' updates by dividing each dimension of the model into $b$ buckets. This enables the servers to build a secret sharing of the frequency distribution of clients' updates and find the bucket which contains the median value. The approximate median is then chosen as the middle value in that bucket.

At the beginning of a given round each client $c_i$ receives bucket range $B$ and global model $\boldsymbol{w}_g$ from the central servers, and follow by running SGD on its dataset $D_i$ to obtain local model $\boldsymbol{w}_i$. The bucket range implicitly fixes the buckets around the global model $\boldsymbol{w}_g$, i.e. in any given dimension $k \in [d]$, the $b$ buckets are centered around $\boldsymbol{w}_g(k)$ with total range of $B$. The ending buckets are taken to represent all the remaining range outside $B$ - i.e. the first and the last buckets represent the range $(-\infty, \boldsymbol{w}_g(k) - \frac{B}{2}], [\boldsymbol{w}_g(k) + \frac{B}{2}, +\infty)$. The middle $b - 2$ buckets evenly divide the range $\boldsymbol{w}_g(k) \pm \frac{B}{2}$. $c_i$ then creates, $\forall k \in [d]$, a unit vector $\boldsymbol{e}_i(k) \in \{0, 1\}^b$ with 1 in place for the bucket which contains $\boldsymbol{w}_i(k)$; we refer to this transformation as "**bucketizing**". $c_i$ then sends secret shares of $e_i$ to both servers

The servers add all the client updates to get a secret-sharing of the frequency distribution of the clients updates and compute the bucket containing the median value for each dimension. This is done by performing a secure comparison to compare each bucket's share of frequency against $\lceil \frac{n}{2} \rceil$. The comparison result is then revealed to learn which bucket contains the median value and the servers take the middle value of that bucket to be the value of the new global model $g'_w$ for that dimension. We refer to this transformation of converting a bucketed model back to a model where each dimension has a numerical value associated with it as "**quantizing**". The bucket range is then updated as $2\|\boldsymbol{w}'_g - \boldsymbol{w}_g\|_1 + p$, where $p$ is a parameter agreed by all parties.

**Complexity** For each dimension, the servers need to only perform $b$ calls to $\mathcal{F}^\ell_{\text{Comp}}$ - hence a total of $db$ calls. Compared to $dn(n-1)/2$ calls for pairwise comparison median computation, we note that if $b$ is a constant $= 10$, then while pairwise comparison median needs number of secure comparisons that grows quadratically with the number of clients, the number of calls in our method is independent of the number of clients.

# 6 Theoretical guarantees

In this section, we theoretically analyze the proposed method in terms of security, robustness, and convergence guarantees.

## 6.1 Robustness and Convergence

We first introduce the notation and settings specific to the robustness and convergence analysis. The proofs in this subsection are provided in section D in the appendix. Given a data distribution $\mathcal{D}$, we assume data samples $\xi \sim \mathcal{D}$ are i.i.d. sampled. Data individual loss function is denoted as $f(\cdot\,;\xi)$ : $\mathbb{R}^d \to \mathbb{R}_+$, i.e., $f(\boldsymbol{w};\xi) \geq 0$ given a parameter $\boldsymbol{w} \in \mathbb{R}^d$. Based on the individual loss function, we denote the population loss function $F = \mathbb{E}_{\xi \sim \mathcal{D}}[f(\cdot\,;\xi)]$, and we assume it is differentiable. Draw $N$ i.i.d. samples for each of the $n$ clients (in total $nN$ samples): $\{\xi_{i,j}\}_{i\in[n],j\in[N]}$. Therefore the client empirical loss function is

$$\hat{F}_i = \frac{1}{N}\sum_{j=1}^{N} f(\cdot\,;\xi_{i,j}) \quad \text{for } i \in [n]. \tag{1}$$

The goal of our setting is to minimize the averaged loss across all clients, i.e., $\arg\min_{\boldsymbol{w}} \quad \frac{1}{n}\sum_{i=1}^{n} \hat{F}_i(\boldsymbol{w})$. In the convergence analysis, we consider gradient descent with step size $\mu$. The range of the buckets at iteration $t$ is denoted as $[-B_t/2, B_t/2]$, and the model parameter are denoted as $\boldsymbol{w}_t \in \mathbb{R}^d$. As detailed in Algorithm 3, we consider the bucket range updated by

$$B_0 \leftarrow p_0, \quad B_t \leftarrow \eta\|\boldsymbol{w}_t - \boldsymbol{w}_{t-1}\|_1 + p_1, \tag{2}$$

where $p_0, p_1, \eta > 0$ are constants. There is a convex parameter set $\mathcal{W} \subseteq \mathbb{R}^d$ such that the parameters during gradient descent for all clients are within $\mathcal{W}$. Given a function $g : \mathbb{R}^d \to \mathbb{R}^d$ and $p, q \in [1, \infty]$, we denote the $(p, q)$-norm of $g$ as $\|g\|_{p,q} = \left(\int_{\mathcal{W}} \|g(\boldsymbol{w})\|_q^p \; \mathrm{d}\mu(\boldsymbol{w})\right)^{1/p}$. In particular, $\|g\|_{\infty,q} = \sup_{\boldsymbol{w}\in\mathcal{W}} \|g(\boldsymbol{w})\|_q$.

The coordinate-wise median operator $\mathrm{Med}(\cdot)$ takes a set of vectors $\{\boldsymbol{v}_i \in \mathbb{R}^d\}_{i=1}^n$ and outputs a vector $\boldsymbol{v} \in \mathbb{R}^d$ such that for each dimension $k$, $\boldsymbol{v}(k)$ is the standard median of $\{\boldsymbol{v}_i(k) \in \mathbb{R}\}_{i=1}^n$. The bucket median is calculated as described in Algorithm 3, and we denote it as $\mathrm{BucketMed}(\cdot)$. Therefore, in the convergence analysis, we consider the global parameter update be

$$\boldsymbol{w}_{t+1} \leftarrow \boldsymbol{w}_t - \mathrm{BucketMed}(\{\mu\nabla\hat{F}_i(\boldsymbol{w}_t)\}_{i\in[n]}; B, b),$$

where $B$ is the bucket range and $b$ is the number of buckets.

We make the following assumptions for the robustness and convergence analysis.

**Assumption 1** ($\beta_\infty$-smoothness). *$F$ is $\beta_\infty$-smooth. Formally, there exists $\beta \geq 0$ such that for $\forall \boldsymbol{w}_1, \boldsymbol{w}_2 \in \mathcal{W}$ we have*

$$\|\nabla F(\boldsymbol{w}_1) - \nabla F(\boldsymbol{w}_2)\|_\infty \leq \beta\|\boldsymbol{w}_1 - \boldsymbol{w}_2\|_1,$$

*Note that $\|\cdot\|_1$ is the dual norm of $\|\cdot\|_\infty$.*

**Assumption 2** (Weak Law of Large Number). *We assume $\nabla F = \mathbb{E}_{\xi\sim\mathcal{D}}[\nabla f(\cdot\,;\xi)]$ exists and the sample mean converges to $\nabla F$ in probability in $\|\cdot\|_{\infty,\infty}$. Formally, let $\xi_j \sim \mathcal{D}$ be i.i.d. sampled, and denote $\delta_{N,\epsilon} := \Pr\left\{\left\|\nabla F - \frac{1}{N}\sum_{j=1}^{N}\nabla f(\cdot\,;\xi_j)\right\|_{\infty,\infty} > \epsilon\right\}$. We assume*

$$\forall \epsilon > 0 : \quad \lim_{N\to\infty} \delta_{N,\epsilon} = 0.$$

Therefore, by definition, $\delta_{N,\epsilon_1} \geq \delta_{N,\epsilon_2}$ if $\epsilon_1 \leq \epsilon_2$. Denote $\epsilon_{N,\delta} = \inf\{\epsilon : \delta_{N,\epsilon} \leq \delta\}$, and we can see that $\epsilon_{N,\delta_1} \geq \epsilon_{N,\delta_2}$ if $\delta_1 \leq \delta_2$. Moreover, we have the following proposition.

**Proposition 6.1.** *With Assumption 2, we have $\forall \delta > 0 : \quad \lim_{N\to\infty} \epsilon_{N,\delta} = 0$.*

With the above assumptions, the following result shows that the proposed bucket median is robust to some extent if ratio of adversarial clients is $\alpha < 1/2$. We provide similar results for dimension-wise median $\mathrm{Med}(\cdot)$ in Theorem D.1 in the appendix. Note that the previous analysis on dimension-wise median in the FL setting (Yin et al., 2018) requires $\alpha$ to be smaller than an easily negative value. We prove the robustness in a new way which holds for any $\alpha < 1/2$.

**Theorem 6.1** (Robustness). *With Assumption 1&2, consider $n_1$ i.i.d. samples $\{\nabla\hat{F}_i\}_{i=1}^{n_1}$ (equation 1) and $n_2$ adversarial vectors $\{v_i \in \mathbb{R}^d\}_{i=1}^{n_2}$. Denote $n := n_1 + n_2$ and $\alpha := n_2/n$ being the ratio of adversaries. Given the bucket range $B > 0$, the number of buckets $b$ and the gradient descent step size $\mu$. If $\alpha < 1/2$ then for $\forall \epsilon > \epsilon_{N,\frac{1-2\alpha}{2(1-\alpha)}}$ with probability at least $1 - (\delta_{N,\epsilon})^{\frac{n}{2}(1-2\alpha)}$, we have*

$$\forall \{v_i \in \mathbb{R}^d\}_{i=1}^{n_2} : \quad \|\text{BucketMed}(\{\mu\nabla\hat{F}_i(w)\}_{i=1}^{n_1} \cup \{v_i\}_{i=1}^{n_2}; B, b) - \mu\nabla F(w)\|_\infty \leq \mu\epsilon + \frac{B}{2b},$$

*for $\forall w \in \mathcal{W}$ satisfying $\mu\|\nabla\hat{F}_i(w)\|_\infty \leq B$ for all $i \in [n_1]$. With the bucket median being robust, we can prove the convergence guarantee for smooth (possibly non-convex) function.*

Note that as $N \to \infty$, both $\delta_{N,\epsilon} \to 0$ and $\epsilon_{N,\delta} \to 0$.

**Theorem 6.2** (Robust Convergence). *With Assumption 1&2, consider $n_1$ normal clients with i.i.d. samples $\{\nabla\hat{F}_i\}_{i=1}^{n_1}$ (equation 1) and $n_2$ faulty clients that can send arbitrary vectors following the proposed protocol. Denote $n := n_1 + n_2$ and $\alpha := n_2/n$ being the ratio of faulty clients. The bucket range adaption is defined by equation 2, where we assume $B_t \leq B$ for all iteration $t$ for a constant $B > 0$, and $p_0 \geq \frac{1}{m\beta}\|\nabla\hat{F}_i(w_0)\|_\infty$ for $\forall i \in [n_1]$. If $\alpha < 1/2$, for $\forall \epsilon > \epsilon_{N,\frac{1-2\alpha}{2(1-\alpha)}}$ and $\forall \epsilon' > 0$, the proposed method with parameters $\mu = \frac{1}{d\beta}$, $p_1 \geq 2\mu\epsilon + \frac{B}{2b}$, $\eta \geq 1 + \beta\mu$ and $b \geq \frac{dB\beta}{2\epsilon'}$ can achieve*

$$\min_{0 \leq t \leq T} \frac{1}{m}\|\nabla F(w_t)\|_2^2 \leq \frac{2\beta F(w_0)}{T+1} + (\epsilon + \epsilon')^2,$$

*with probability at least $1 - (\delta_{N,\epsilon})^{\frac{n}{2}(1-2\alpha)}$.*

Thus, with sufficient number of buckets $b$ and number of data $N$, the proposed method can converge w.r.t. the true loss function $F$ with high probability, while $\alpha < 1/2$ of clients send incorrect local updates.

## 6.2 Overview of Security Analysis

We argue the security of our bucketed protocol by means of the standard real/ideal simulation paradigm in cryptography (defined formally in Appendix B). As stated in section 4.1, we assume a semi-honest adversary statically corrupting up to $n-1$ clients and at most one server, and prove the security of one epoch of the FL training using the simulation paradigm. The security of the entire protocol then follows by applying the sequential composition theorem to the security of each epoch.

We provide a brief overview of the security argument. The use of secure 2-party computation between the servers allows them to compute the desired approximate median without revealing anything other than the input/output of the functionality. At the beginning of each round, the clients secret-share their inputs between the two servers and thereafter, the two servers use secure 2-party protocols to compute the shares of a unit vector with 1 for the bucket in which the median value lies. Revealing this unit vector allows the servers to learn the median bucket index and find the median value, which is then sent to clients. An adversary corrupting one of the servers only sees random shares throughout the protocol (guaranteed by the secure protocol implementing $\mathcal{F}_{\text{Comp}}^\ell, \mathcal{F}_{\text{Equal}}^\ell$, described in Appendix C) in addition to bucket range (which is computed solely using public values) and median bucket index, which it sees in the clear. Additionally corrupting any number of clients does not allow an adversary to recover any honest client's information, since honest client's data are only secret-shared between the servers and not shared in plain between clients.

We revisit the simulation paradigm and provide the formal security proof in Appendix B.

## 7 Experimental Results

In this section, we evaluate our proposed bucketed median scheme on different models and datasets. First, we compare convergence behaviour over epoch of FL of bucketed median, pairwise comparison median, and mean methods. We also detail the quadratic versus linear relationship the pairwise comparison and bucketed median methods respectively have with the number of clients. In the next subsection, we formally describe the failures that faulty clients can exhibit. We then compare the robustness of median and our bucketed median against mean. In Appendix F, we detail a breakdown of one FL cycle and analyze sensitivity of parameters.

Experiments were conducted on an Intel(R) Xeon(R) Platinum 8280 CPU @ 2.70GHz virtual machine with 112 cores. All parties involved in the FL process were separate processes on this machine,

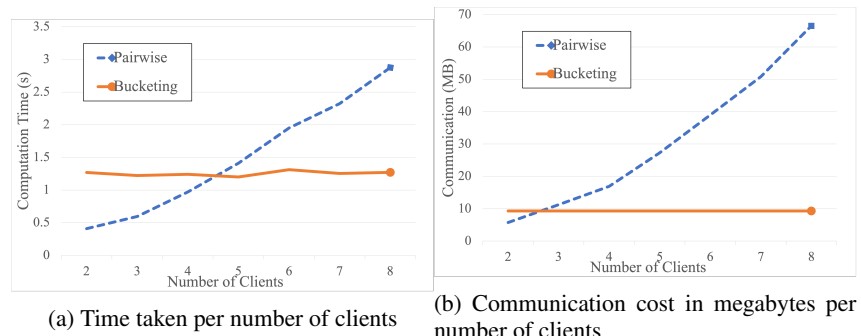

(a) Time taken per number of clients

(b) Communication cost in megabytes per number of clients

Figure 3: Single-threaded median computation over 1000 dimensions.

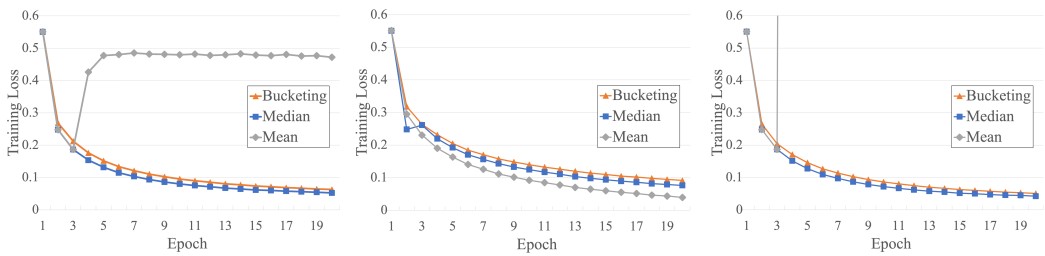

(a) MLP, bitflip failure     (b) MLP, label flip failure     (c) MLP, Gaussian noise failure

Figure 4: Training loss convergence behaviour for 3 clients FL system using different server-side methods.

with a bandwidth cap of 377 MBps to simulate a LAN network. We test on the following combinations of (model, dataset): (MLP, MNIST), (CNNMnist, Mnist), and (CNN, CIFAR-10). The MLP model has consits of 1 hidden layer with 200 units with ReLu activation for $199, 210$ dimensions. We use the same CNNMnist as McMahan et al. (2017), which has 1,663,370 dimensions. The CNN model we use consists of two $5 \times 5$ convolution layers, the first followed by a $2 \times 2$ max pooling, and the second followed by three linear layers for a total of 581,866 dimensions. Model training was done using the CPU. Training batch size is set to 20, learning rate is set to $0.01$, and client datasets are sampled i.i.d. from the global dataset. Number of buckets $b = 8$.

## 7.1 Convergence Results

We first test convergence behaviour when all parties are benign. We show that our bucketed median FL system exhibits similar training loss convergence as our baseline: a mean based FL system. For all three methods, we run the same number of iterations of FL and note the average training loss when all clients are benign.

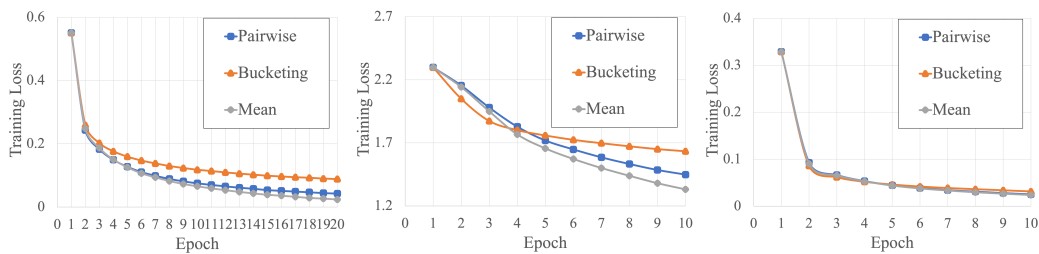

(a) 20 iter. of (MLP, MNIST)    (b) 10 iter. of (CNN, CIFAR-10)    (c) 10 iter. of (CNNMnist, MNIST)

Figure 2: Results of training loss convergence on 3 benign parties.

We also show the relationship between number of parties and the time taken for secure median computation in both the bucketing and pairwise median methods. We test for number of clients $n \in [2, 10]$ for dimension-wise median when $d = 1000$. Figure 3 depicts the constant and quadratic relationships between time and communication costs with number of clients for both bucketing and pairwise methods. In Appendix F, we provide scalability results with larger number of clients.

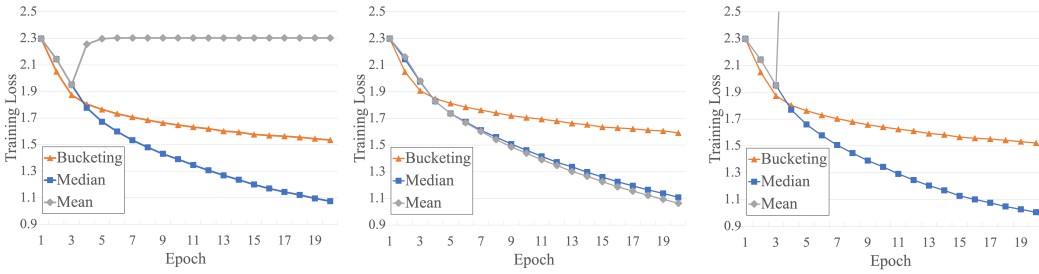

(a) CNN, bitflip failure     (b) CNN, label flip failure     (c) CNN, Gaussian noise failure

Figure 5: Training loss convergence behaviour for 3 clients FL system using different server-side methods.

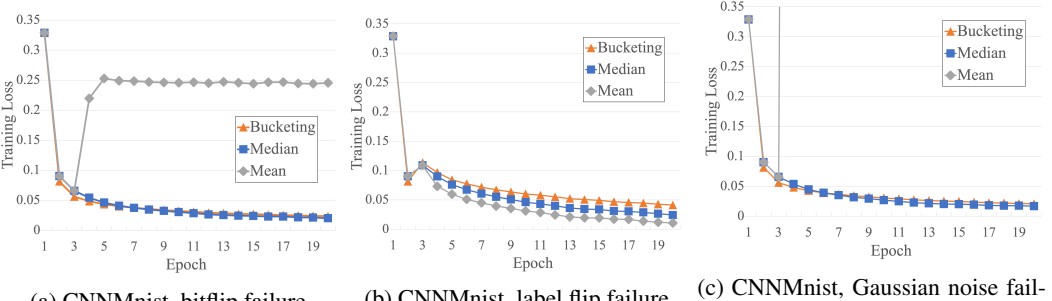

(a) CNNMnist, bitflip failure     (b) CNNMnist, label flip failure     (c) CNNMnist, Gaussian noise failure

Figure 6: Training loss convergence behaviour for 3 clients FL system using different server-side methods.

## 7.2 Client Failure Types

We assume the updates sent from clients can be corrupted with errors arising from machine noise, network transmission errors, data poisoning etc. The failures we test with are:

- **Bit Flip Failure:** The most significant bit, or the bit that controls the sign is flipped. The client sends the negative gradient instead of the true gradient to the servers.
- **Label Switching Failure:** Clients compute gradients with switched labels on the training data. For example, $label \in \{0, ..., 9\}$ is replaced by $9 - label$.
- **Gaussian Noise Failure:** Instead of sending the correct gradient to the server, clients send an update with values sampled from a Gaussian distribution with mean of zero and standard deviation of 200.

## 7.3 Robustness Results

In this subsection, we empirically show that the robustness of our bucketed median FL system behaves similarly to a median based FL system. We show that for bit-flip, label-flip, and Gaussian noise errors, our system still converges. We test with three clients, one of which exhibits failures starting from the third epoch. Figures $4, 5, 6$ shows results of our bucketed median method versus a median and mean method. In our implementation, the faulty client performs label flipping during the training process, while bitflip and Gaussian noise are manually computed on the local model returned from model training. The mean method results in loss of convergence for both bitflip and Gaussian noise failures, but performs well against label flip failures. However, both the median and bucketed median methods show continued training loss decrease against all three failures.

## 8 Conclusion

We presented a dual central-server FL system that is robust and private in the semi-honest setting. We propose a bucketing approximation technique in order to sacrifice accuracy for computational efficiency, which also leads to better scalability with the number of clients. The advantage our work provides are both privacy and robustness in distributed ML.

## Ethics Statement

Our work focuses on theoretically and empirically studying robust federated learning. All the datasets and packages we use are open-sourced. Potential ethical concerns are primarily in the observation that robust federated learning tends to remove outliers, and it can be hard to determine if data from minority groups are outliers or adversarial.

## Reproducibility Statement

We have tried our best to provide training details to facilitate reproducing our results. We provide detailed results on how to train and evaluate our model. We will open-source our code once accepted.

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

# A  Full Description of our algorithms

Here we provide full descriptions of both our median based and approximate bucketed median based FL algorithms (Algorithm 2, Algorithm 3), along with more details and correctness sketch for the same.

## A.1  Median based FL system

This is formally specified in full in Algorithm 2.

**Correctness sketch.**  We sketch the idea behind the protocol, which is inspired from similar ideas used in (Tueno et al., 2019). To compute the median of $n$ values - $\{x_1, x_2 \ldots x_n\}$, we first compute all pairwise comparison results - $b_{ij} = (x_i > x_j), \forall i, j \in [n], i \neq j$. This allows us to compute for each element $x_i$, its position in the sorted list of $\{x_1, x_2 \ldots x_n\}$, by computing $pos_i = \sum_{k \in [n], k \neq i} b_{ik}$. Finally, we find element for which $pos_i = \lceil n/2 \rceil$.

In our actual computation, we only compute $b_{ij}, \forall i, j \in [n]$ s.t. $j > i$. Then $b_{ji}$ is taken as $1 - b_{ij}$. The correctness of this median depends on all the values being different, which we assume to happen in our protocol (and abort if this is not true).

**Float-to-fixed conversion.**  Our protocol for exact median involves mapping the client's local floating point values to integers, since it is well known that most cryptographic protocols are much more efficient when working over integer rings than directly over floating point values (Mohassel & Zhang, 2017; Mohassel & Rindal, 2018; Kumar et al., 2020). Previous work in this area (Mohassel & Zhang, 2017; Mohassel & Rindal, 2018; Kumar et al., 2020), has made use of fixed point representation for the same, where real number $r$ is mapped to integer $\rho_r = \lfloor r * 2^s \rfloor$, where $s$ is a parameter called the scaling factor. Hence, given a real number $r$, we map it to its fixed point representation $\rho_r$, which is then mapped to an integer ring like $\mathbb{Z}_{2^\ell}$. In the following, we use $w_i^{\text{Int}}$ to denote fixed point version of $w_i$.

F2I$_{\ell,s}$ and I2F$_{\ell,s}$ are functions which are used for the mapping real numbers to (fixed-point) integers and vice versa respectively. In particular, F2I$_{\ell,s} : \mathbb{R} \to \mathbb{Z}_{2^\ell}$, is defined as F2I$_{\ell,s}(x) = \lfloor x * 2^s \rfloor$. I2F$_{\ell,s} : \mathbb{Z}_{2^\ell} \to \mathbb{R}$ is defined as I2F$_{\ell,s}(x) = x/2^s$ (where $x \in \mathbb{Z}_{2^\ell}$ is first interpreted as a signed integer and then the division is performed).

Since we use $\ell = 64, s = 24$ and 32-bit floating point values throughout and which have only 23 bits of precision (and since we are only performing addition over the fixed point values/the values involved at any intermediate stage are small) we don't suffer any accuracy loss/convergence rate loss due to the usage of fixed point values.

## A.2  Bucketed median based FL

In Algorithm 3 we describe in full our algorithm to compute the bucketed median.

---

**Algorithm 2** One training epoch of exact median based FL

Input: $\forall i \in [n]$, $c_i$ holds dataset $D_i$

Parameters: Bitlength $\ell$, fixed point scale $s$

---

$\forall i \in [n]$, client $c_i$ does:

1. Receive fixed-point shares of global model $\langle \boldsymbol{w}_g^{\text{Int}} \rangle_j \in \mathbb{Z}_{2^\ell}$ from server $S_j, \forall j \in \{0, 1\}$.

2. Compute $\boldsymbol{w}_g^{\text{Int}} = \langle \boldsymbol{w}_g^{\text{Int}} \rangle_0 + \langle \boldsymbol{w}_g^{\text{Int}} \rangle_1$ and $\boldsymbol{w}_g = \text{I2F}_{\ell,s}(\boldsymbol{w}_g^{\text{Int}})$.

3. Compute local model $\boldsymbol{w}_i = \text{SGD}(\boldsymbol{w}_g, D_i)$.

4. Convert back to fixed point as $\boldsymbol{w}_i^{\text{Int}} = \text{F2I}_{\ell,s}(\boldsymbol{w}_i)$.

5. Create fresh $(2, 2)$ secret shares of $\boldsymbol{w}_i^{\text{Int}}$ by choosing $\langle \boldsymbol{w}_i^{\text{Int}} \rangle_0$ at random from $\mathbb{Z}_{2^\ell}^d$ and computing $\langle \boldsymbol{w}_i^{\text{Int}} \rangle_1 = \boldsymbol{w}_i^{\text{Int}} - \langle \boldsymbol{w}_i^{\text{Int}} \rangle_0$.

6. $\forall j \in \{0, 1\}$, send $\langle \boldsymbol{w}_i^{\text{Int}} \rangle_j$ to server $S_j$.

$\forall j \in \{0, 1\}$, server $S_j$ performs the following:

1. $\forall i \in [n]$, receive share $\langle \boldsymbol{w}_i^{\text{Int}} \rangle_j$ from client $c_i$.

2. $\forall k \in [d]$, compute shares of median in dimension $j$ across all the $n$ clients as follows:

   (a) $\forall x \in [n], y \in [x + 1, n]$, compute $\langle z_{x,y}^k \rangle_j$ where $z_{x,y}^k = (\boldsymbol{w}_x^{\text{Int}}(k) > \boldsymbol{w}_y^{\text{Int}}(k))$ by calling $\mathcal{F}_{\text{Comp}}^\ell$ with inputs $\langle \boldsymbol{w}_x^{\text{Int}}(k) \rangle_j, \langle \boldsymbol{w}_y^{\text{Int}}(k) \rangle_j$. Set $\langle z_{y,x}^k \rangle_j = j - \langle z_{x,y}^k \rangle_j$.

   (b) $\forall i \in [n]$, compute $\langle pos_i \rangle_j = \sum_{x \in [n], x \neq i} \langle z_{i,x}^k \rangle_j$.

   (c) Set $mid = (n + 1)/2$ if $n$ is odd, else set $mid = n/2$.

   (d) $\forall i \in [n]$, check if $pos_i = mid$ by calling $\mathcal{F}_{\text{Equal}}^\ell$ with inputs $\langle pos_i \rangle_j, j * mid$ and receiving output $\langle eq_i \rangle_j$ in return.

   (e) $\forall i \in [n]$, send $\langle eq_i \rangle_j$ to server $S_{1-j}$ and compute $eq_i = \langle eq_i \rangle_0 + \langle eq_i \rangle_1$.

   (f) Find $i \in [n]$ s.t. $eq_i(k) = 1$ and set $\langle \boldsymbol{w}_g^{\text{Int}'}(k) \rangle_j = \langle \boldsymbol{w}_i^{\text{Int}}(k) \rangle_j$.

3. Send $\langle \boldsymbol{w}_g^{\text{Int}'} \rangle_j$ to each client $c_i, \forall i \in [n]$.

---

---

**Algorithm 3** One training epoch of bucketed median based FL

Input: $\forall i \in [n]$, clients $c_i$ holds dataset $D_i$

Parameters: Number of buckets $b$, bitlength $\ell$, total epochs $r$, constant values $p_0, p_1, p_t = \frac{p_1}{t}$, where $t$ is the current epoch number.

---

[h] $\forall i \in [n]$, client $c_i$ does:

1. Receive global model $\boldsymbol{w}_g$ and bucket range $B$ from server $S_0$.

2. Compute local model $\boldsymbol{w}_i = \text{SGD}(\boldsymbol{w}_g, D_i)$.

3. **Bucketize** $\boldsymbol{w}_i$ to get $\boldsymbol{e}_i \in \{0,1\}^{d \times b}$. Specifically, for each dimension $k$ in $[d]$, compute the $k^{th}$ row of bucketed model $\boldsymbol{e}_i$ as follows:

   - If $\boldsymbol{w}_i(k) \leq \boldsymbol{w}_g(k) - \frac{B}{2}$, set $\boldsymbol{e}_i(k)$ as $[1, 0, \ldots, 0]$.
   - If $\boldsymbol{w}_i(k) \geq \boldsymbol{w}_g(k) + \frac{B}{2}$, set $\boldsymbol{e}_i(k)$ as $[0, \ldots, 0, 1]$.
   - Else, let $z = \lfloor \frac{\boldsymbol{w}_i(k) - (\boldsymbol{w}_g(k) - B/2)}{B/(b-2)} \rfloor + 1$. Set $\boldsymbol{e}_i(k)$ as $[0, \ldots, 0, 1, 0, \ldots, 0]$, where $\boldsymbol{e}_i(k, z)$ is set to 1.

4. Create $(2,2)$ secret shares of $\boldsymbol{e}_i$ by choosing $\langle \boldsymbol{e}_i \rangle_0$ at random from $\mathbb{Z}_{2^\ell}^{d \times b}$ and computing $\langle \boldsymbol{e}_i \rangle_1 = \boldsymbol{e}_i - \langle \boldsymbol{e}_i \rangle_0$.

5. $\forall j \in \{0,1\}$, send $\langle \boldsymbol{e}_i \rangle_j$ to server $S_j$.

$\forall j \in \{0,1\}$, server $S_j$ performs the following:

1. $\forall i \in [n]$, receive share $\langle \boldsymbol{e}_i \rangle_j$ from client $c_i$.

2. Set $\langle \boldsymbol{h} \rangle_j = \sum_{i \in [n]} \langle \boldsymbol{e}_i \rangle_j$.

3. For each dimension $k$ in $[d]$, calculate the approximate median:

   (a) $\forall y \in [2, b]$, set $\langle \boldsymbol{h}(k, y) \rangle_j = \sum_{z \in [y]} \langle \boldsymbol{h}(k, z) \rangle_j$.

   (b) Find the bucket that contains the median:

        i. Compute $\langle \boldsymbol{m}(k) \rangle_j$, where $\boldsymbol{m}(k)$ is of form $[0, \ldots, 1, \ldots, 0]$, by calling $\mathcal{F}_{\text{Comp}}^\ell$ with $S_0$'s input being $\langle \boldsymbol{h}(k) \rangle_0$, $[\lceil \frac{n}{2} \rceil]^{d \times b}$ and $S_1$'s input being $\langle \boldsymbol{h}(k) \rangle_1$, $[0]^{d \times b}$. Set $\langle \boldsymbol{m}(k) \rangle_j = [j]^b - \langle \boldsymbol{m}(k) \rangle_j$.

        ii. $S_j$ shares $\langle \boldsymbol{m} \rangle_j$ with $S_{1-j}$.

        iii. $S_0$ computes $\boldsymbol{m} = \langle \boldsymbol{m}(k) \rangle_0 \bigoplus \langle \boldsymbol{m}(k) \rangle_1$.

        iv. $S_0$ selects the bucket $y$ that contains the median as $\min_{y \in [0, b-1)} [\boldsymbol{m}(k, y) \equiv 1]$.

   (c) $S_0$ **quantizes** the approximate median value $\boldsymbol{w}_g'(k)$ as follows:

   - If $y == 0$, $\boldsymbol{w}_g'(k) = \boldsymbol{w}_g(k) - \frac{B}{2}$
   - If $y == b - 1$, $\boldsymbol{w}_g'(k) = \boldsymbol{w}_g(k) + \frac{B}{2}$
   - Else, $\boldsymbol{w}_g'(k) = \left( (\boldsymbol{w}_g(k) - \frac{B}{2} + (y-1) * (\frac{B}{b-2})) + (\boldsymbol{w}_g(k) - \frac{B}{2} + y * (\frac{B}{b-2})) \right) / 2$.

4. $S_0$ creates new bucket range $B' = 2\|\boldsymbol{w}_g' - \boldsymbol{w}_g\|_1 + p_{1,t}$, where $p_{1,t}$ is predetermined.

5. $S_0$ sends $\boldsymbol{w}_g'$, $B'$ to client $c_i, \forall i \in [n]$.

To initialize FL, server $S_0$ performs the following:

1. Initialize $\boldsymbol{w}_g$ as model with parameter values of 0.

2. Set $B = p_0$.

3. $S_0$ sends $\boldsymbol{w}_g$, $B$ to each client $c_i, \forall i \in [n]$.

---

# B  Formal security Proof

In this work, we use the semi-honest or honest-but-curious adversarial model, where it is assumed that the adversary follows the protocol specifications, but tries to glean information about the other party's inputs using the protocol transcript. As stated previously, we assume a semi-honest adversary, statically (i.e. before starting the protocol) corrupting up to $n-1$ clients and one of the servers, and prove full security of the protocol using the simulation paradigm.

### B.1 Defining Secure Multiparty Computation

Here, we provide a formal definition of secure multiparty computation. Parts of this section have been borrowed verbatim from (Goldreich, 2004).

A multi-party protocol is cast by specifying a random process that maps pairs of inputs to pairs of outputs (one for each party). We refer to such a process as a functionality. The security of a protocol is defined with respect to a functionality $f$. In particular, let $n$ denote the number of parties. A non-reactive $n$-party functionality $f$ is a (possibly randomized) mapping of $n$ inputs to $n$ outputs. A multiparty protocol with security parameter $\lambda$ for computing a non-reactive functionality $f$ is a protocol running in time polynomial in $\lambda$ and satisfying the following correctness requirement: if parties $P_1, \ldots, P_n$ with inputs $(x_1, \ldots, x_n)$ respectively all run an honest execution of the protocol, then the joint distribution of the outputs $y_1, \ldots, y_n$ of the parties is statistically close to $f(x_1, \ldots, x_n)$. A reactive functionality $f$ is a sequence of non-reactive functionalities $f = (f_1, \ldots, f_\ell)$ computed in a stateful fashion in a series of phases. Let $x_i^j$ denote the input of $P_i$ in phase $j$, and let $s^j$ denote the state of the computation after phase $j$. Computation of $f$ proceeds by setting $s^0$ equal to the empty string and then computing $(y_1^j, \ldots, y_n^j, s^j) \leftarrow f_j(s^{j-1}, x_1^j, \ldots, x_n^j)$ for $j \in [\ell]$, where $y_i^j$ denotes the output of $P_i$ at the end of phase $j$. A multi-party protocol computing $f$ also runs in $\ell$ phases, at the beginning of which each party holds an input and at the end of which each party obtains an output. (Note that parties may wait to decide on their phase-$j$ input until the beginning of that phase.) Parties maintain state throughout the entire execution. The correctness requirement is that, in an honest execution of the protocol, the joint distribution of all the outputs $\{y_1^j, \ldots, y_n^j\}_{j=1}^{\ell}$ of all the phases is statistically close to the joint distribution of all the outputs of all the phases in a computation of $f$ on the same inputs used by the parties.

**Defining Security.** The security of a protocol (with respect to a functionality $f$) is defined by comparing the real-world execution of the protocol with an ideal-world evaluation of $f$ by a trusted party. More concretely, it is required that for every adversary $\mathcal{A}$, which attacks the real execution of the protocol, there exist an adversary Sim, also referred to as a simulator, which can *achieve the same effect* in the ideal-world. Next, we specify what each of these worlds mean.

**The real execution** In the real execution of the n-party protocol $\pi$ for computing $f$ is executed in the presence of an adversary $\mathcal{A}$. The honest parties follow the instructions of $\pi$. The adversary $\mathcal{A}$ takes as input the security parameter $k$, the set $I \subset [n]$ of corrupted parties, the inputs of the corrupted parties, and an auxiliary input $z$. $\mathcal{A}$ obtains the local randomness and the state of all corrupted parties, but all corrupted parties do follow the strategy outlined in the protocol specification.

The interaction of $\mathcal{A}$ with a protocol $\pi$ defines a random variable $\mathsf{REAL}_{\pi, \mathcal{A}(z), I}(\lambda, \vec{x})$ whose value is determined by the coin tosses of the adversary and the honest players. This random variable contains the output of the adversary (which may be an arbitrary function of its view, which includes the state of all corrupted parties) as well as the outputs of the uncorrupted parties. We let $\mathsf{REAL}_{\pi, \mathcal{A}(z), I}$ denote the distribution ensemble $\{\mathsf{REAL}_{\Pi, \mathcal{A}(z), I}(\lambda, \vec{x})\}_{k \in \mathsf{N}, \langle \vec{x}, z \rangle \in \{0,1\}^*}$.

**The ideal execution – security with abort.** In this second variant of the ideal model, fairness and output delivery are no longer guaranteed. This is the standard relaxation used when a strict majority of honest parties is not assumed. In this case, an ideal execution for a function $f$ proceeds as follows:

- **Send inputs to the trusted party:** As before, the parties send their inputs to the trusted party, and we let $x_i'$ denote the value sent by $P_i$. Once again, for a semi-honest adversary we require $x_i' = x_i$ for all $i \in I$.

- **Trusted party sends output to the adversary:** The trusted party computes $f(x_1', \ldots, x_n') = (y_1, \ldots, y_n)$ and sends $\{y_i\}_{i \in I}$ to the adversary.

- **Adversary instructs trusted party to abort or continue:** This is formalized by having the adversary send either a continue or abort message to the trusted party. (A semi-honest adversary never aborts.) In the latter case, the trusted party sends to each uncorrupted party $P_i$ its output value $y_i$. In the former case, the trusted party sends the special symbol $\perp$ to each uncorrupted party.

- **Outputs:** Sim outputs an arbitrary function of its view, and the honest parties output the values obtained from the trusted party.

The interaction of Sim with the trusted party defines a random variable $\mathsf{IDEAL}_{f,\mathcal{A}(z)}(\lambda, \vec{x})$ as above,and we let $\{\mathsf{IDEAL}_{f,\mathcal{A}(z),I}(\lambda, \vec{x})\}_{k \in \mathsf{N}, \langle \vec{x}, z \rangle \in \{0,1\}^*}$. Having defined the real and the ideal worlds, we now proceed to define our notion of security, where a semi-honest adversary is defined as an (interactive) Turing Machine whose specifications match those of the protocol.

**Definition 1.** *Let $k$ be the security parameter. Let $f$ be an $n$-party randomized functionality, and $\Pi$ be an $n$-party protocol for $n \in \mathsf{N}$. We say that $\Pi$ $t$-securely computes $f$ in the presence of semi-honest adversaries if for every semi-honest adversary $\mathcal{A}$ there exists a semi-honest adversary $\mathsf{Sim}$ such that for any $I \subset [n]$ with $|I| \le t$ the following quantity is negligible:*

$$|Pr[\mathsf{REAL}_{\Pi,\mathcal{A}(z),I}(\lambda, \vec{x}) = 1] - Pr[\mathsf{IDEAL}_{f,\mathcal{A}(z),I}(\lambda, \vec{x}) = 1]|$$

*where $\vec{x} = \{x_i\}_{i \in [n]} \in \{0,1\}^*$ and $z \in \{0,1\}^*$.*

### B.2 Simulation

Since our model is semi-honest and our protocols use secure 2-party computation, the simulator in our case is straightforward. Given the inputs of the adversary and the bucket index containing the median value across clients, the simulator sends random shares to the corrupted server on behalf of the honest clients. Finally, during secure bucketed median computation, it invokes the simulator for the secure protocol implementing $\mathcal{F}_{\mathsf{Comp}}^{\ell}$, and to this simulator it provides as input the bucket index of the median value.

## C  Protocols implementing $\mathcal{F}_{\mathsf{Comp}}^{\ell}, \mathcal{F}_{\mathsf{Equal}}^{\ell}$

For exposition purposes, we described our protocols for median computation by writing them in a "hybrid" model, where parties have access to trusted functionalities $\mathcal{F}_{\mathsf{Comp}}^{\ell}, \mathcal{F}_{\mathsf{Equal}}^{\ell}$, computing secret sharing of comparison and equality of two inputs respectively (described in Figure 7 and Figure 8 respectively). In this section, we provide protocols implementing the said functionalities. Our final protocol is then obtained by composing these with our own protocols. We then provide reasoning on why we choose secure comparisons as the metric for protocol complexity.

---

$\mathcal{F}_{\mathsf{Comp}}^{\ell}$

Parties - $P_0, P_1$
- $\forall b \in \{0,1\}, P_b$ sends $x_b, y_b \in \mathbb{Z}_{2^\ell}$ to $\mathcal{F}_{\mathsf{Comp}}^{\ell}$.
- $\mathcal{F}_{\mathsf{Comp}}^{\ell}$ computes $x = x_0 + x_1 \in \mathbb{Z}_{2^\ell}, y = y_0 + y_1 \in \mathbb{Z}_{2^\ell}$ and bit $z = (x > y)$.
- $\mathcal{F}_{\mathsf{Comp}}^{\ell}$ randomly chooses $z_0 \in \mathbb{Z}_{2^\ell}$ and computes $z_1 = z - z_0 \in \mathbb{Z}_{2^\ell}$ and sends $z_b$ to $P_b, \forall b \in \{0,1\}$.

---

Figure 7: Ideal functionality for 2-party comparisons: given $\langle x \rangle, \langle y \rangle$ compute $\langle x > y \rangle$.

---

$\mathcal{F}_{\mathsf{Equal}}^{\ell}$

Parties - $P_0, P_1$
- $\forall b \in \{0,1\}, P_b$ sends $x_b, y_b \in \mathbb{Z}_{2^\ell}$ to $\mathcal{F}_{\mathsf{Equal}}^{\ell}$.
- $\mathcal{F}_{\mathsf{Equal}}^{\ell}$ computes $x = x_0 + x_1 \in \mathbb{Z}_{2^\ell}, y = y_0 + y_1 \in \mathbb{Z}_{2^\ell}$ and bit $z = (x = y)$.
- $\mathcal{F}_{\mathsf{Equal}}^{\ell}$ randomly chooses $z_0 \in \mathbb{Z}_{2^\ell}$ and computes $z_1 = z - z_0 \in \mathbb{Z}_{2^\ell}$ and sends $z_b$ to $P_b, \forall b \in \{0,1\}$.

---

Figure 8: Ideal functionality for 2-party equality: given $\langle x \rangle, \langle y \rangle$ compute $\langle x = y \rangle$.

**Notation**   Before going further, lets setup some notation. For a boolean variable $z$, we use notation $\mathbf{1}\{z\}$ to indicate the indicator variable, which is $1$ iff $z$ is true. While $\langle x \rangle_0, \langle x \rangle_1$ denote $(2,2)$ additive secret shares of $x$ over $\mathbb{Z}_{2^\ell}$, we use notation $\langle x \rangle_0^B, \langle x \rangle_1^B$ to denote the $(2,2)$ additive secret shares of $x$ over $\mathbb{Z}_2$.

To implement $\mathcal{F}_{\mathsf{Comp}}^{\ell}, \mathcal{F}_{\mathsf{Equal}}^{\ell}$, we use and modify a bit the protocols proposed in CrypTFlow2 (Rathee et al., 2020). We describe the protocols for these in the following hybrids:

- $\mathcal{F}_{\text{Mill}}^{\ell}$: This is described in Figure 9. It computes the solution to the millionaire's problem over $\mathbb{Z}_{2^{\ell}}$ and provides boolean secret shares of the answer bit to the two parties.

- $\mathcal{F}_{\text{B2A}}^{\ell}$: This is described in Figure 10. It takes in boolean shares of a bit from the two parties and provides shares of the same over $\mathbb{Z}_{2^{\ell}}$.

- $\mathcal{F}_{\text{Mill,eq}}^{\ell}$: This is described in Figure 11. It is similar to $\mathcal{F}_{\text{Mill}}^{\ell}$, except instead of doing comparison, it checks equality of the two received values.

Each of the above hybrids can be implemented directly using protocols from CrypTFlow2 (Rathee et al., 2020). In more detail, while protocols for $\mathcal{F}_{\text{Mill}}^{\ell}, \mathcal{F}_{\text{B2A}}^{\ell}$ are described directly in (Rathee et al., 2020), the protocol for $\mathcal{F}_{\text{Mill,eq}}^{\ell}$ can be obtained directly by a small tweak[1] to the protocol for $\mathcal{F}_{\text{Mill}}^{\ell}$. Since these protocols follow directly from (Rathee et al., 2020), we skip providing detailed descriptions and proofs of them here and refer the reader to CrypTFlow2 (Rathee et al., 2020) for the same.

Given the above three hybrids, we provide the protocols for implementing $\mathcal{F}_{\text{Comp}}^{\ell}, \mathcal{F}_{\text{Equal}}^{\ell}$ in Algorithm 4, Algorithm 5 respectively. We then have the following theorems (where "securely" refers to simulation-based security according to Definition 1).

**Theorem C.1.** $\Pi_{Comp}^{\ell}$ securely implements $\mathcal{F}_{Comp}^{\ell}$ in the $\mathcal{F}_{Mill}^{\ell-1}, \mathcal{F}_{B2A}^{\ell}$ hybrid as long as the signed inputs of the two parties $x, y$ (whose secret-shares they input into the protocol) satisfy $|x| + |y| < 2^{\ell-1}$.

**Theorem C.2.** $\Pi_{Equal}^{\ell}$ securely implements $\mathcal{F}_{Equal}^{\ell}$ in the $\mathcal{F}_{Mill,eq}^{\ell}, \mathcal{F}_{B2A}^{\ell}$ hybrid.

Composing the corresponding implementation of $\mathcal{F}_{\text{Mill}}^{\ell}, \mathcal{F}_{\text{B2A}}^{\ell}, \mathcal{F}_{\text{Mill,eq}}^{\ell}$ (as described above), gives us secure protocols for $\mathcal{F}_{\text{Comp}}^{\ell}, \mathcal{F}_{\text{Equal}}^{\ell}$.

---

$\mathcal{F}_{\textbf{Mill}}^{\ell}$

Parties - $P_0, P_1$
- $P_0$ sends $x \in \{0,1\}^{\ell}$ to $\mathcal{F}_{\text{Mill}}^{\ell}$, while $P_1$ sends $y \in \{0,1\}^{\ell}$ to $\mathcal{F}_{\text{Mill}}^{\ell}$.
- $\mathcal{F}_{\text{Mill}}^{\ell}$ computes $z = (x < y)$, chooses $\langle z \rangle_0^B \leftarrow \{0,1\}$ at random and computes $\langle z \rangle_1^B = \langle z \rangle_0^B \oplus z$. $\mathcal{F}_{\text{Mill}}^{\ell}$ sends $\langle z \rangle_b^B$ to $P_b, \forall b \in \{0,1\}$.

Figure 9: Ideal functionality for solving the Millionaire's problem

---

$\mathcal{F}_{\textbf{B2A}}^{\ell}$

Parties - $P_0, P_1$
- $\forall b \in \{0,1\}, P_b$ sends $\langle x \rangle_b^B$ to $\mathcal{F}_{\text{B2A}}^{\ell}$.
- $\mathcal{F}_{\text{B2A}}^{\ell}$ computes $x = \langle x \rangle_0^B \oplus \langle x \rangle_1^B$, chooses $\langle x \rangle_0 \leftarrow \mathbb{Z}_{2^{\ell}}$ at random and computes $\langle x \rangle_1 \in \mathbb{Z}_{2^{\ell}}$ as $\langle x \rangle_1 = x - \langle x \rangle_0$. It sends $\langle x \rangle_b$ to $P_b, \forall b \in \{0,1\}$.

Figure 10: Ideal functionality for converting boolean to arithmetic shares

---

[1](Rathee et al., 2020) provides protocols for $\mathcal{F}_{\text{Mill}}^{\ell}$ and $\mathcal{F}_{\text{B2A}}^{\ell}$ directly. The protocol for $\mathcal{F}_{\text{Mill}}^{\ell}$ works by breaking up the $\ell$ bits of inputs into a balanced binary tree and computing at each level the answer of both comparison and equality checks of the inputs corresponding to that subtree. Hence, while not stated directly in the protocol in (Rathee et al., 2020), a protocol for $\mathcal{F}_{\text{Mill,eq}}^{\ell}$ can be easily obtained from the protocol for $\mathcal{F}_{\text{Mill}}^{\ell}$. In addition, the public implementation of CrypTFlow2 (cry, 2021), contains functions which allow us to compute the same without tweaking anything underlying.

---

$$\mathcal{F}^{\ell}_{\text{Mill,eq}}$$

Parties - $P_0, P_1$
- $P_0$ sends $x \in \{0,1\}^{\ell}$ to $\mathcal{F}^{\ell}_{\text{Mill,eq}}$, while $P_1$ sends $y \in \{0,1\}^{\ell}$ to $\mathcal{F}^{\ell}_{\text{Mill,eq}}$.
- $\mathcal{F}^{\ell}_{\text{Mill,eq}}$ computes $z = (x = y)$, chooses $\langle z \rangle^B_0 \leftarrow \{0,1\}$ at random and computes $\langle z \rangle^B_1 = \langle z \rangle^B_0 \oplus z$. $\mathcal{F}^{\ell}_{\text{Mill,eq}}$ sends $\langle z \rangle^B_b$ to $P_b, \forall b \in \{0,1\}$.

Figure 11: Ideal functionality checking if two parties hold the same value

---

**Algorithm 4** $\ell$ bit signed comparison protocol, $\Pi^{\ell}_{\text{Comp}}$.

---

**Input**: $\forall b \in \{0,1\}, P_b$ holds $\langle x \rangle_b, \langle y \rangle_b$.
**Output**: $\forall b \in \{0,1\}, P_b$ holds $\langle \mathbf{1}\{x > y\} \rangle_b$

1. $\forall b \in \{0,1\}, P_b$ computes $\langle z \rangle_b = \langle x \rangle_b - \langle y \rangle_b$.
2. Compute boolean shares of $msb(z)$ as follows:
   (a) $\forall b \in \{0,1\}, P_b$ parses $\langle z \rangle_b$ as $msb_b || y_b$.
   (b) $P_0$ and $P_1$ invoke $\mathcal{F}^{\ell-1}_{\text{Mill}}$ with $P_0$'s input $2^{\ell-1} - y_0$ and $P_1$'s input $y_1$. $\forall b \in \{0,1\}, P_b$ learns $\langle carry \rangle^B_b$.
   (c) $\forall b \in \{0,1\}, P_b$ computes $\langle t \rangle^B_b = msb_b \oplus \langle carry \rangle^B_b$.
3. $P_0$ and $P_1$ invoke $\mathcal{F}^{\ell}_{\text{B2A}}$ with inputs $\langle t \rangle^B_b$ to learn $\langle t \rangle_b$.

---

**Algorithm 5** $\ell$ bit equality protocol, $\Pi^{\ell}_{\text{Equal}}$.

---

**Input**: $\forall b \in \{0,1\}, P_b$ holds $\langle x \rangle_b, \langle y \rangle_b$.
**Output**: $\forall b \in \{0,1\}, P_b$ holds $\langle \mathbf{1}\{x = y\} \rangle_b$

1. $\forall b \in \{0,1\}, P_b$ computes $\langle z \rangle_b = \langle x \rangle_b - \langle y \rangle_b$.
2. Check if $z = 0$ by doing the following:
   (a) $P_0$ and $P_1$ invoke computes $\mathcal{F}^{\ell}_{\text{Mill,eq}}$ with $P_0$'s input $\langle z \rangle_0$, while $P_1$'s input being $(-\langle z \rangle_1) \bmod 2^{\ell}$. $\forall b \in \{0,1\}, P_b$ learns $\langle eq \rangle^B_b$.
   (b) $P_0$ and $P_1$ invoke $\mathcal{F}^{\ell}_{\text{B2A}}$ with inputs $\langle eq \rangle^B_b$ to learn $\langle eq \rangle_b$.

---

### C.1 Why we use secure comparison as complexity metric

MPC is efficient on linear operations when using arithmetic shares. The sum of two arithmetic shares $\langle a \rangle + \langle b \rangle$ result in share $\langle a + b \rangle$. However, this is not the case for non-linear operations such as $\langle a \rangle > \langle b \rangle$, since the operation performed and the type of secret sharing are not of the same nature.

These type of non-linear operations such as multiplication, comparisons, and equality require increased computations, such as generation of correlated randomness between parties. There have been many works on improving the efficiency of secure comparisons. (Damgård et al., 2006) provide a secure comparison protocol which uses $O(\ell \log \ell)$ secure multiplications, where $\ell = \log p$ and $p$ is the prime used for the field the secret shares are computed over. (Nishide & Ohta, 2007) improved the complexity of secure comparison to $O(\ell)$ secure multiplications.

Thus a large portion of the evaluation of a function in MPC is due to the computation of non-linear operations. Computing median intuitively involves expensive comparison operations, and therefore we choose the number of secure comparisons needed as a metric for computational complexity of the protocols.

## D  Detailed Robustness and Convergence Analysis

**Proposition D.1** (Proposition 6.1 Restated). *With Assumption 2, we have*

$$\forall \delta > 0 : \quad \lim_{N \to \infty} \epsilon_{N,\delta} = 0.$$

*Proof.* Given a $\delta > 0$, for $\forall \epsilon > 0$, by Assumption 2 there exists $N$ such that $\forall M : M > N$ we have $\delta_{M,\epsilon} < \delta/2$. Therefore, $\epsilon_{M,\delta} \leq \epsilon_{M,\delta/2} = \inf\{\epsilon' : \delta_{M,\epsilon'} \leq \delta/2\} \leq \epsilon$. $\qquad\square$

**Theorem D.1.** *With Assumption 1&2, consider $n_1$ i.i.d. samples $\{\nabla \hat{F}_i\}_{i=1}^{n_1}$ (equation 1) and $n_2$ adversarial vectors $\{\boldsymbol{v}_i \in \mathbb{R}^d\}_{i=1}^{n_2}$. Denote $n := n_1 + n_2$ and $\alpha := n_2/n$ being the ratio of adversaries. If $\alpha < 1/2$, then for $\forall \epsilon > \epsilon_{N, \frac{1-2\alpha}{2(1-\alpha)}}$ with probability at least $1 - (\delta_{N,\epsilon})^{\frac{n}{2}(1-2\alpha)}$, we have*

$$\forall \{\boldsymbol{v}_i \in \mathbb{R}^d\}_{i=1}^{n_2} : \quad \|\mathrm{Med}(\{\nabla \hat{F}_i(\boldsymbol{w})\}_{i=1}^{n_1} \cup \{\boldsymbol{v}_i\}_{i=1}^{n_2}) - \nabla F(\boldsymbol{w})\|_\infty \leq \epsilon,$$

*for $\forall \boldsymbol{w} \in \mathcal{W}$. Note that as $N \to \infty$, both $\delta_{N,\epsilon} \to 0$ and $\epsilon_{N,\delta} \to 0$.*

*Proof.* By definition of the median operation, we have

$$\|\mathrm{Med}(\{\nabla \hat{F}_i(\boldsymbol{w})\}_{i=1}^{n_1} \cup \{\boldsymbol{v}_i\}_{i=1}^{n_2}) - \nabla F(\boldsymbol{w})\|_\infty$$
$$= \|\mathrm{Med}(\{\nabla \hat{F}_i(\boldsymbol{w}) - \nabla F(\boldsymbol{w})\}_{i=1}^{n_1} \cup \{\boldsymbol{v}_i - \nabla F(\boldsymbol{w})\}_{i=1}^{n_2})\|_\infty. \tag{3}$$

Invoking Lemma E.2, we have

$$(3) \leq \mathrm{Med}(\{\|\nabla \hat{F}_i(\boldsymbol{w}) - \nabla F(\boldsymbol{w})\|_\infty\}_{i=1}^{n_1} \cup \{\|\boldsymbol{v}_i - \nabla F(\boldsymbol{w})\|_\infty\}_{i=1}^{n_2})$$
$$= \mathrm{Med}(\{\|\nabla \hat{F}_i(\boldsymbol{w}) - \nabla F(\boldsymbol{w})\|_\infty\}_{i=1}^{n_1} \cup \{a_i\}_{i=1}^{n_2}), \tag{4}$$

where $a_i = \|\boldsymbol{v}_i - \nabla F(\boldsymbol{w})\|_\infty \in \mathbb{R}$.

As we can see, (4) depends on the variance of $\nabla \hat{F}_i = \frac{1}{N} \sum_{j=1}^{N} \nabla f(\cdot\,; \xi_{i,j})$, where $\xi_{i,j} \sim \mathcal{D}$ are i.i.d. sampled. Denote

$$\hat{X}_{i,\epsilon} := \mathbf{1}\{\|\nabla F - \nabla \hat{F}_i\|_{\infty,\infty} \leq \epsilon\},$$

where $\mathbf{1}(\cdot)$ is the indicator function. By Assumption 2, we can see that $\hat{X}_i$ is a Bernoulli random variable with

$$\Pr\left\{\hat{X}_{i,\epsilon} = 1\right\} = 1 - \delta_{N,\epsilon}.$$

Therefore, $\hat{X}_\epsilon := \sum_{i=1}^{n_1} \hat{X}_{i,\epsilon}$ forms a binomial distribution. By the Chernoff bound for binomial distribution, for $k \in \mathbb{N}$ and $k \leq n_1(1 - \delta_{N,\epsilon})$, we have

$$\Pr(\hat{X}_\epsilon \leq k) \leq e^{-n_1 D_{KL}\left(\frac{k}{n_1} \left\| (1 - \delta_{N,\epsilon})\right.\right)}, \tag{5}$$

where $D_{KL}$ is the Kullback–Leibler divergence, i.e., for $q_1, q_2 \in (0,1)$

$$D_{KL}(q_1 \| q_2) := q_1 \log\left(\frac{q_1}{q_2}\right) + (1 - q_1) \log\left(\frac{1 - q_1}{1 - q_2}\right)$$
$$\geq q_1 \log q_1 + (1 - q_1) \log(1 - q_1) - (1 - q_1) \log(1 - q_2)$$
$$\geq (1 - q_1) \log \frac{1}{1 - q_2},$$

where the first inequality is due to that $q_2 < 1$ and the last inequality is because that $q_1 \log q_1 + (1 - q_1) \log(1 - q_1)$ is the entropy which is non-negative.

Combining the above inequality with equation 5, we have

$$\Pr(\hat{X}_\epsilon \leq k) \leq e^{-n_1(1 - \frac{k}{n_1}) \log \frac{1}{\delta_{N,\epsilon}}} = (\delta_{N,\epsilon})^{n_1 - k}. \tag{6}$$

Denoting event

$$E_\epsilon := \{\hat{X}_\epsilon > \frac{n_1 + n_2}{2}\}, \tag{7}$$

we can see that if $E_\epsilon$ happens then by Lemma E.1 we have $(4) \leq \epsilon$. It left to show the probability of event $E_\epsilon$ happens. Therefore, when $\frac{n_1+n_2}{2n_1} \leq (1 - \delta_{N,\epsilon})$, we can apply equation 6 to have

$$\Pr(E_\epsilon) = 1 - \Pr(\hat{X}_\epsilon \leq \frac{n_1 + n_2}{2}) \geq 1 - (\delta_{N,\epsilon})^{\frac{n}{2}(1-2\alpha)}. \tag{8}$$

The condition of $\frac{n_1+n_2}{2n_1} \leq (1 - \delta_{N,\epsilon})$ is equivalent to $\delta_{N,\epsilon} \leq \frac{1-2\alpha}{2(1-\alpha)}$, and it has a sufficient condition of $\epsilon > \epsilon_{N, \frac{1-2\alpha}{2(1-\alpha)}}$.

$\qquad\square$

**Theorem D.2** (Theorem 6.1 Restated). *With Assumption 1&2, consider $n_1$ i.i.d. samples $\{\nabla \hat{F}_i\}_{i=1}^{n_1}$ (equation 1) and $n_2$ adversarial vectors $\{\boldsymbol{v}_i \in \mathbb{R}^d\}_{i=1}^{n_2}$. Denote $n := n_1 + n_2$ and $\alpha := n_2/n$ being the ratio of adversaries. Given the bucket range $B > 0$, the number of buckets $b$ and the gradient descent step size $\mu$. If $\alpha < 1/2$ then for $\forall \epsilon > \epsilon_{N, \frac{1-2\alpha}{2(1-\alpha)}}$ with probability at least $1 - (\delta_{N,\epsilon})^{\frac{n}{2}(1-2\alpha)}$, we have*

$$\forall \{\boldsymbol{v}_i \in \mathbb{R}^d\}_{i=1}^{n_2} : \quad \|\mathrm{BucketMed}(\{\mu\nabla\hat{F}_i(\boldsymbol{w})\}_{i=1}^{n_1} \cup \{\boldsymbol{v}_i\}_{i=1}^{n_2}; B, b) - \mu\nabla F(\boldsymbol{w})\|_\infty \le \mu\epsilon + \frac{B}{2b},$$

*for $\forall \boldsymbol{w} \in \mathcal{W}$ satisfying $\mu\|\nabla\hat{F}_i(\boldsymbol{w})\|_\infty \le B$ for all $i \in [n]$.*

*Proof.* First, by the definition of $\mathrm{BucketMed}(\cdot)$, for any input vector $\boldsymbol{v}_i \in \mathbb{R}^d$ to the $\mathrm{BucketMed}(\cdot)$, there exists a vector $\boldsymbol{v}_i' \in \mathbb{R}^d$ with $\|\boldsymbol{v}_i'\|_\infty \le B$ such that

$$\mathrm{BucketMed}(\{\mu\nabla\hat{F}_i(\boldsymbol{w})\}_{i=1}^{n_1} \cup \{\boldsymbol{v}_i\}_{i=1}^{n_2}; B, b) = \mathrm{BucketMed}(\{\mu\nabla\hat{F}_i(\boldsymbol{w})\}_{i=1}^{n_1} \cup \{\boldsymbol{v}_i'\}_{i=1}^{n_2}; B, b).$$

Denote $E_\epsilon$ as the event defined in equation 7 By equation 8 we can see that

$$\Pr(E_\epsilon) \ge 1 - (\delta_{N,\epsilon})^{\frac{n}{2}(1-2\alpha)}.$$

Given the event $E_\epsilon$, by Theorem D.1 we can see that

$$\|\mathrm{Med}(\{\mu\nabla\hat{F}_i(\boldsymbol{w})\}_{i=1}^{n_1} \cup \{\boldsymbol{v}_i'\}_{i=1}^{n_2}) - \mu\nabla F(\boldsymbol{w})\|_\infty \le \mu\epsilon, \tag{9}$$

for $\forall \boldsymbol{w} \in \mathcal{W}$. Moreover, for $\forall \boldsymbol{w} \in \mathcal{W}$ satisfying $\mu\|\nabla\hat{F}_i(\boldsymbol{w})\|_\infty \le B$ for all $i \in [n_1]$, by Lemma E.3 we have

$$\|\mathrm{BucketMed}(\{\mu\nabla\hat{F}_i(\boldsymbol{w})\}_{i=1}^{n_1} \cup \{\boldsymbol{v}_i'\}_{i=1}^{n_2}; B, b) - \mathrm{Med}(\{\mu\nabla\hat{F}_i(\boldsymbol{w})\}_{i=1}^{n_1} \cup \{\boldsymbol{v}_i'\}_{i=1}^{n_2})\|_\infty \le \frac{B}{2b}. \tag{10}$$

Therefore, combining equation 9 and equation 10 gives the theorem. $\square$

**Theorem D.3** (Theorem 6.2 Restated). *With Assumption 1&2, consider $n_1$ normal clients with i.i.d. samples $\{\nabla\hat{F}_i\}_{i=1}^{n_1}$ (equation 1) and $n_2$ faulty clients that can send arbitrary vectors following the proposed protocol. Denote $n := n_1 + n_2$ and $\alpha := n_2/n$ being the ratio of faulty clients. The bucket range adaption is defined by equation 2, where we assume $B_t \le B$ for all iteration $t$ for a constant $B > 0$, and $p_0 \ge \frac{1}{m\beta}\|\nabla\hat{F}_i(\boldsymbol{w}_0)\|_\infty$ for $\forall i \in [n_1]$. If $\alpha < 1/2$, for $\forall\epsilon > \epsilon_{N, \frac{1-2\alpha}{2(1-\alpha)}}$ and $\forall\epsilon' > 0$, the proposed method with parameters $\mu = \frac{1}{d\beta}$, $p_1 \ge 2\mu\epsilon + \frac{B}{2b}$, $\eta \ge 1 + \beta\mu$ and $b \ge \frac{dB\beta}{2\epsilon'}$ can achieve*

$$\min_{0 \le t \le T} \frac{1}{m}\|\nabla F(\boldsymbol{w}_t)\|_2^2 \le \frac{2\beta F(\boldsymbol{w}_0)}{T+1} + (\epsilon + \epsilon')^2,$$

*with probability at least $1 - (\delta_{N,\epsilon})^{\frac{n}{2}(1-2\alpha)}$.*

*Proof.* Denote $E_\epsilon := \{\hat{X}_\epsilon > \frac{n_1+n_2}{2}\}$ as the event defined in equation 7, where

$$\hat{X}_\epsilon := \sum_{i=1}^{n_1} \mathbf{1}\{\|\nabla F - \nabla\hat{F}_i\|_{\infty,\infty} \le \epsilon\}.$$

By equation 8 we have

$$\Pr(E_\epsilon) \ge 1 - (\delta_{N,\epsilon})^{\frac{n}{2}(1-2\alpha)}.$$

Now condition on the event $E_\epsilon$, and denote the set of clients

$$I := \{i \in [n_1] : \|\nabla F - \nabla\hat{F}_i\|_{\infty,\infty} \le \epsilon\}, \tag{11}$$

and we can see that $|I| > \frac{n_1+n_2}{2}$.

Recall that the training procedure of Algorithm 3 is

$$\boldsymbol{w}_t = \boldsymbol{w}_{t-1} - \mathrm{BucketMed}(\{\mu\nabla\hat{F}_i(\boldsymbol{w}_{t-1})\}_{i=1}^{n_1} \cup \{\boldsymbol{v}_{i,t-1}\}_{i=1}^{n_2}; B_{t-1}, b),$$

where $\boldsymbol{v}_{i,t-1}$ are sent from the faulty clients $i$ at iteration $t$. In the following, for notational ease we denote

$$\boldsymbol{g}_{t-1} := \mathrm{BucketMed}(\{\mu\nabla\hat{F}_i(\boldsymbol{w}_{t-1})\}_{i=1}^{n_1} \cup \{\boldsymbol{v}_{i,t-1}\}_{i=1}^{n_2}; B_{t-1}, b).$$

Moreover recall the bucket range $B_t$ at iteration $t$ is updated by

$$B_t = \eta\|\boldsymbol{w}_t - \boldsymbol{w}_{t-1}\|_1 + p_1 = \eta\|\boldsymbol{g}_{t-1}\|_1 + p_1.$$

Now, given any $i \in I$, we make the following claim.

**Claim 1.** *Conditioned on event $E_\epsilon$, for any iteration step $t \geq 0$ we have*

$$\mu\|\nabla\hat{F}_i(\boldsymbol{w}_t)\|_\infty \leq B_t, \tag{12}$$

$$\|\mu\nabla F(\boldsymbol{w}_t) - \boldsymbol{g}_t\|_\infty \leq \mu\epsilon + \frac{B}{2b}. \tag{13}$$

*Proof of Claim 1.* Note that equation 12 is true for $t = 0$ by assumption, and we prove the claim by induction. Assuming equation 12 holds for iteration $t - 1$, we prove equation 12 for iteration $t$, and prove equation 13 for iteration $t - 1$ by the way. We begin by

$$\mu\|\nabla\hat{F}_i(\boldsymbol{w}_t)\|_\infty$$
$$=\mu\|\nabla\hat{F}_i(\boldsymbol{w}_t) - \nabla F(\boldsymbol{w}_t) + \nabla F(\boldsymbol{w}_t) - \nabla F(\boldsymbol{w}_{t-1}) + \nabla F(\boldsymbol{w}_{t-1}) - \boldsymbol{g}_{t-1}/\mu + \boldsymbol{g}_{t-1}/\mu\|_\infty$$
$$\leq \underbrace{\mu\|\nabla\hat{F}_i(\boldsymbol{w}_t) - \nabla F(\boldsymbol{w}_t)\|_\infty}_{a_1} + \underbrace{\mu\|\nabla F(\boldsymbol{w}_t) - \nabla F(\boldsymbol{w}_{t-1})\|_\infty}_{a_2}$$
$$+ \underbrace{\|\mu\nabla F(\boldsymbol{w}_{t-1}) - \boldsymbol{g}_{t-1}\|_\infty}_{a_3} + \underbrace{\|\boldsymbol{g}_{t-1}\|_\infty}_{a_4}, \tag{14}$$

where the last step is by triangle inequality. We will show that $a_1, a_2, a_4$ have straight-forward upper bounds, respectively. For the $a_1$, by equation 11 we have

$$a_1 \leq \mu\epsilon.$$

By Assumption 1, we have

$$a_2 \leq \mu\beta\|\boldsymbol{w}_t - \boldsymbol{w}_{t-1}\|_1 = \mu\beta\|\boldsymbol{g}_{t-1}\|_1.$$

For $a_4$, by definition of the $\ell_1$ norm and $\ell_\infty$ norm, we directly have

$$a_4 \leq \|\boldsymbol{g}_{t-1}\|_1.$$

Note that $a_3$ is exactly we want to prove for equation 13. For $a_3$, we need to prove a slightly different version of Theorem 6.1, however, using similar techniques. Recall the definition of $\boldsymbol{g}_{t-1}$ and $I \subseteq [n_1]$, we have

$$a_3 = \|\mu\nabla F(\boldsymbol{w}_{t-1}) - \text{BucketMed}(\{\mu\nabla\hat{F}_i(\boldsymbol{w}_{t-1})\}_{i=1}^{n_1} \cup \{\boldsymbol{v}_{i,t-1}\}_{i=1}^{n_2}; B_{t-1}, b)\|_\infty$$

$$= \|\mu\nabla F(\boldsymbol{w}_{t-1}) - \text{BucketMed}(\{\mu\nabla\hat{F}_i(\boldsymbol{w}_{t-1})\}_{i\in I} \cup \{\mu\nabla\hat{F}_i(\boldsymbol{w})\}_{i\in[n_1]\setminus I} \cup \{\boldsymbol{v}_{i,t-1}\}_{i=1}^{n_2}; B_{t-1}, b)\|_\infty.$$

Note that we have assumed that $\|\mu\nabla\hat{F}_i(\boldsymbol{w}_{t-1})\| \leq B_{t-1}$. For $\{\mu\nabla\hat{F}_i(\boldsymbol{w})\}_{i\in[n_1]\setminus I} \cup \{\boldsymbol{v}_{i,t-1}\}_{i=1}^{n_2}$, there exists $n_1 + n_2 - |I|$ vectors $\boldsymbol{v}_i' \in \mathbb{R}^d$ satisfying $\|\boldsymbol{v}_i'\|_\infty \leq B_{t-1}$ for $i \in [n_1 + n_2 - |I|]$ such that

$$\text{BucketMed}(\{\mu\nabla\hat{F}_i(\boldsymbol{w}_{t-1})\}_{i=1}^{n_1} \cup \{\boldsymbol{v}_{i,t-1}\}_{i=1}^{n_2}; B_{t-1}, b)$$
$$= \text{BucketMed}(\{\mu\nabla\hat{F}_i(\boldsymbol{w}_{t-1})\}_{i\in I} \cup \{\mu\nabla\hat{F}_i(\boldsymbol{w})\}_{i\in[n_1]\setminus I} \cup \{\boldsymbol{v}_{i,t-1}\}_{i=1}^{n_2}; B_{t-1}, b)$$
$$= \text{BucketMed}(\{\mu\nabla\hat{F}_i(\boldsymbol{w}_{t-1})\}_{i\in I} \cup \{\boldsymbol{v}_i'\}_{i=1}^{n_1+n_2-|I|}; B_{t-1}, b) \tag{15}$$

Therefore, denoting $\mathcal{V} := \{\mu\nabla\hat{F}_i(\boldsymbol{w}_{t-1})\}_{i\in I} \cup \{\boldsymbol{v}_i'\}_{i=1}^{n_1+n_2-|I|}$, by Lemma E.3, we have

$$\|\text{BucketMed}(\mathcal{V}; B_{t-1}, b) - \text{Med}(\mathcal{V})\|_\infty \leq \frac{B_{t-1}}{2b} \leq \frac{B}{2b}. \tag{16}$$

Moreover, since $|I| > \frac{n_1+n_2}{2}$ and $\|\nabla F(\boldsymbol{w}_{t-1}) - \nabla\hat{F}(\boldsymbol{w}_{t-1})\|_\infty \leq \epsilon$ for any $i \in I$, by Lemma E.1 we have

$$\|\text{Med}(\mathcal{V}) - \nabla F(\boldsymbol{w}_{t-1})\|_\infty = \|\text{Med}(\{\mu\nabla\hat{F}_i(\boldsymbol{w}_{t-1})\}_{i\in I} \cup \{\boldsymbol{v}_i'\}_{i=1}^{n_1+n_2-|I|}) - \nabla F(\boldsymbol{w}_{t-1})\|_\infty$$
$$\leq \mu\epsilon. \tag{17}$$

Combining equation 16 and equation 17 by triangle inequality, we have

$$\|\text{BucketMed}(\mathcal{V}; B_{t-1}, b) - \mu\nabla F(\boldsymbol{w}_{t-1})\|_\infty \leq \mu\epsilon + \frac{B}{2b}.$$

Noting that $\mathcal{V} = \{\mu\nabla\hat{F}_i(\boldsymbol{w}_{t-1})\}_{i\in I} \cup \{\boldsymbol{v}_i'\}_{i=1}^{n_1+n_2-|I|}$, we apply equation 15 to have

$$\|\text{BucketMed}(\{\mu\nabla\hat{F}_i(\boldsymbol{w}_{t-1})\}_{i=1}^{n_1} \cup \{\boldsymbol{v}_{i,t-1}\}_{i=1}^{n_2}; B_{t-1}, b) - \mu\nabla F(\boldsymbol{w}_{t-1})\|_\infty \leq \mu\epsilon + \frac{B}{2b},$$

which proves equation 13 for iteration $t - 1$, and equivalently $a_3 \leq \mu\epsilon + \frac{B}{2b}$. Collecting our upper bounds for $a_1, a_2, a_3$ and $a_4$, we finally have

$$(14) = a_1 + a_2 + a_3 + a_4 \leq \mu\epsilon + \mu\beta\|\boldsymbol{g}_{t-1}\|_1 + \mu\epsilon + \frac{B}{2b} + \|\boldsymbol{g}_{t-1}\|_1$$

$$= 2\mu\epsilon + \frac{B}{2b} + (1 + \mu\beta)\|\boldsymbol{g}_{t-1}\|_1.$$

Recall the condition that $p_1 \geq 2\mu\epsilon + \frac{B}{2b}$ and $\eta \geq 1 + \beta\mu$. We can see that

$$(14) \leq \eta\|\boldsymbol{g}_{t-1}\|_1 + p_1 = B_t.$$

By induction, it concludes the claim.

$\square$

Given the Claim 1, we continue the proof of Theorem 6.2. By the smoothness assumption (Assumption 1), and given that the parameter domain $\mathcal{W}$ is defined to be convex, it is known that

$F(\boldsymbol{w}_{t+1}) - F(\boldsymbol{w}_t)$

$$\leq \langle \nabla F(\boldsymbol{w}_t), \boldsymbol{w}_{t+1} - \boldsymbol{w}_t \rangle + \frac{\beta}{2}\|\boldsymbol{w}_{t+1} - \boldsymbol{w}_t\|_1^2$$

$$= -\langle \nabla F(\boldsymbol{w}_t), \boldsymbol{g}_t \rangle + \frac{\beta}{2}\|\boldsymbol{g}_t\|_1^2$$

$$\leq -\langle \nabla F(\boldsymbol{w}_t), \boldsymbol{g}_t \rangle + \frac{d\beta}{2}\|\boldsymbol{g}_t\|_2^2$$

$$= -\langle \nabla F(\boldsymbol{w}_t), \boldsymbol{g}_t - \mu\nabla F(\boldsymbol{w}_t) + \mu\nabla F(\boldsymbol{w}_t) \rangle + \frac{d\beta}{2}\|\boldsymbol{g}_t - \mu\nabla F(\boldsymbol{w}_t) + \mu\nabla F(\boldsymbol{w}_t)\|_2^2$$

$$= -\mu\|\nabla F(\boldsymbol{w}_t)\|_2^2 - \langle \nabla F(\boldsymbol{w}_t), \boldsymbol{g}_t - \mu\nabla F(\boldsymbol{w}_t) \rangle$$

$$+ \frac{d\beta}{2}(\mu^2\|\nabla F(\boldsymbol{w}_t)\|_2^2 + \|\boldsymbol{g}_t - \mu\nabla F(\boldsymbol{w}_t)\|_2^2 + \mu\langle \nabla F(\boldsymbol{w}_t), \boldsymbol{g}_t - 2\mu\nabla F(\boldsymbol{w}_t) \rangle).$$

$$(18)$$

Substituting into $\mu = \frac{1}{d\beta}$, we can continue as

$$(18) = -\frac{1}{2d\beta}\|\nabla F(\boldsymbol{w}_t)\|_2^2 + \frac{d\beta}{2}\|\boldsymbol{g}_t - \mu\nabla F(\boldsymbol{w}_t)\|_2^2$$

$$\leq -\frac{1}{2d\beta}\|\nabla F(\boldsymbol{w}_t)\|_2^2 + \frac{d\beta}{2}\|\boldsymbol{g}_t - \mu\nabla F(\boldsymbol{w}_t)\|_\infty^2.$$

By Claim 1, we have

$$(18) \leq -\frac{1}{2d\beta}\|\nabla F(\boldsymbol{w}_t)\|_2^2 + \frac{d\beta}{2}\left(\mu\epsilon + \frac{B}{2b}\right)^2$$

$$= -\frac{1}{2d\beta}\|\nabla F(\boldsymbol{w}_t)\|_2^2 + \left(\frac{\epsilon}{\sqrt{2\beta}} + \sqrt{\frac{\beta}{2}}\frac{dB}{2b}\right)^2.$$

Therefore, rearranging the inequality, and denote

$$\rho := 2\beta\left(\frac{\epsilon}{\sqrt{2\beta}} + \sqrt{\frac{\beta}{2}}\frac{dB}{2b}\right)^2 = \left(\epsilon + \frac{dB\beta}{2b}\right)^2,$$

we have

$$\frac{1}{d}\|\nabla F(\boldsymbol{w}_t)\|_2^2 \leq 2\beta(F(\boldsymbol{w}_t) - F(\boldsymbol{w}_{t+1})) + \rho.$$

Accordingly, for number of iterations $T \geq 1$:

$$\frac{1}{T+1}\sum_{t=0}^{T}\frac{1}{d}\|\nabla F(\boldsymbol{w}_t)\|_2^2 \leq \frac{2\beta}{T+1}(F(\boldsymbol{w}_0) - F(\boldsymbol{w}_{T+1})) + \rho$$

$$\leq \frac{2\beta F(\boldsymbol{w}_0)}{T+1} + \rho. \qquad (19)$$

Given any $\epsilon' > 0$ and choosing $b > \frac{dB\beta}{2\epsilon'}$, equation 19 implies

$$\min_{0 \leq t \leq T}\frac{1}{d}\|\nabla F(\boldsymbol{w}_t)\|_2^2 \leq \frac{2\beta F(\boldsymbol{w}_0)}{T+1} + (\epsilon + \epsilon')^2.$$

$\square$

# E  Auxiliary Lemmas

**Lemma E.1.** *Given a set of real number $\mathcal{A} = \{x_i \in \mathbb{R}\}_{i=1}^n$ and an interval $[a, b] \subset \mathbb{R}$, if $|\mathcal{A} \cap [a, b]| > |\mathcal{A} \cap [a, b]^c|$ then we have $\mathrm{Med}(\mathcal{A}) \in [a, b]$.*

*Proof.* Without loss of generality, we assume $\mathcal{A}$ is sorted, i.e, $x_i \leq x_{i+1}$ for $\forall i \in [n-1]$. By the definition of median, there are at least $\lceil \frac{n}{2} \rceil$ numbers in $\mathcal{A}$ that are less or equal to $\mathrm{Med}(\mathcal{A})$. Therefore, if $\mathrm{Med}(\mathcal{A}) < a$, we can see that there are at least $\lceil \frac{n}{2} \rceil$ numbers in $\mathcal{A}$ that are outside $[a, b]$, which contradicts to the condition of $|\mathcal{A} \cap [a, b]| > |\mathcal{A} \cap [a, b]^c|$. Hence, $\mathrm{Med}(\mathcal{A}) \geq a$. Similarly, we can show $\mathrm{Med}(\mathcal{A}) \leq b$. That being said, $\mathrm{Med}(\mathcal{A}) \in [a, b]$. □

**Lemma E.2.** *Given $n$ vectors $\{\boldsymbol{v}_i \in \mathbb{R}^d\}_{i=1}^n$, we have*

$$\|\mathrm{Med}(\{\boldsymbol{v}_i\}_{i=1}^n)\|_\infty \leq \mathrm{Med}(\{\|\boldsymbol{v}_i\|_\infty\}_{i=1}^n).$$

*Proof.* Denoting $\boldsymbol{v}_i(k)$ as the $k^{th}$ dimension of $\boldsymbol{v}_i \in \mathbb{R}^d$, we have

$$\|\mathrm{Med}(\{\boldsymbol{v}_i\}_{i=1}^n)\|_\infty = \max_{k \in [m]} |\mathrm{Med}(\{\boldsymbol{v}_i(k)\}_{i=1}^n)| \leq \max_{k \in [m]} \mathrm{Med}(\{|\boldsymbol{v}_i(k)|\}_{i=1}^n). \tag{20}$$

Denote $k^\star \in \arg\max_{k \in [m]} \mathrm{Med}(\{|\boldsymbol{v}_i(k)|\}_{i=1}^n)$, and we have

$$(20) = \mathrm{Med}(\{|\boldsymbol{v}_i(k^\star)|\}_{i=1}^n) \leq \mathrm{Med}(\{\max_{k \in [m]} |\boldsymbol{v}_i(k)|\}_{i=1}^n) = \mathrm{Med}(\{\|\boldsymbol{v}_i\|_\infty\}_{i=1}^n).$$

□

**Lemma E.3.** *Given the bucket range $B > 0$, the number of buckets $b$, for any $n$ vectors $\{\boldsymbol{v}_i\}_{i=1}^n$ in $\mathbb{R}^d$, if $\|\boldsymbol{v}_i\|_\infty \leq B$ for all $i \in [n]$, then we have*

$$\|\mathrm{BucketMed}(\{\boldsymbol{v}_i\}_{i=1}^n; B, b) - \mathrm{Med}(\{\boldsymbol{v}_i\}_{i=1}^n)\|_\infty \leq \frac{B}{2b}.$$

*Proof.* Since $\|\boldsymbol{v}_i\|_\infty \leq B$ for all $i \in [n]$ are within the bucket range, the $\mathrm{BucketMed}(\{\boldsymbol{v}_i\}_{i=1}^n; B, b)$ is the same as $\mathrm{Med}(\{\boldsymbol{v}_i'\}_{i=1}^n)$ for some $\boldsymbol{v}_i' \in \mathbb{R}^d$ satisfying

$$\|\boldsymbol{v}_i' - \boldsymbol{v}_i\|_\infty \leq \frac{B}{2b}.$$

Denote

$$\boldsymbol{w}_i^l = \boldsymbol{v}_i - \frac{B}{2b}, \qquad \boldsymbol{w}_i^r = \boldsymbol{v}_i + \frac{B}{2b},$$

where the $\pm \frac{B}{2b}$ is done on every dimension of $\boldsymbol{v}$. We can see that $\boldsymbol{w}_i^l \leq \boldsymbol{v}_i' \leq \boldsymbol{w}_i^r$, where the $\leq$ is also dimension-wise. Noting that

$$\mathrm{Med}(\{\boldsymbol{v}_i\}_{i=1}^n) - \frac{B}{2b} = \mathrm{Med}(\{\boldsymbol{w}_i^l\}_{i=1}^n) \leq \mathrm{Med}(\{\boldsymbol{v}_i'\}_{i=1}^n) \leq \mathrm{Med}(\{\boldsymbol{w}_i^r\}_{i=1}^n) = \mathrm{Med}(\{\boldsymbol{v}_i\}_{i=1}^n) + \frac{B}{2b},$$

we can conclude that

$$\|\mathrm{Med}(\{\boldsymbol{v}_i'\}_{i=1}^n) - \mathrm{Med}(\{\boldsymbol{v}_i\}_{i=1}^n)\| \leq \frac{B}{2b}.$$

Replacing the $\mathrm{Med}(\{\boldsymbol{v}_i'\}_{i=1}^n)$ by $\mathrm{BucketMed}(\{\boldsymbol{v}_i\}_{i=1}^n; B, b)$ gives the lemma. □

# F  Computation Breakdown and Parameter Sensitivity

In this subsection, we detail a temporal breakdown of a round of federated learning in our approach and the pairwise comparison median. We show the relationship between server-side computation time and increasing number of clients. In the bucketing technique, parameters are agreed upon at the start of the FL protocol to initiate values for the bucketing ranges. We empirically show that these parameters are not sensitive, and that convergence still holds while varying the order of magnitude of these parameter values.

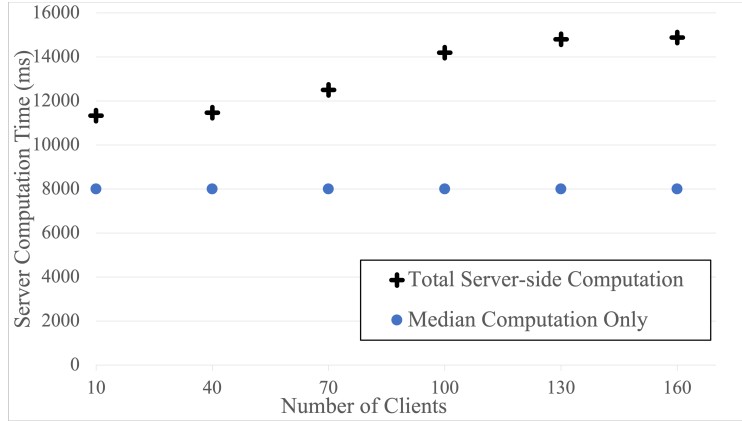

Figure 12: Relationship between number of clients and 1. total server-side computation time after receiving all client updates, 2. bucketed median computation time.

| Computation Breakdown for one round of FL | | | | | |
|---|---|---|---|---|---|
| **Insecure Median** | Time (s) | **Pairwise Comparison Median** | Time (s) | **Bucketed Median** | Time (s) |
| Clients train local model | 24 | Clients train local model | 24 | Clients train local model | 24 |
| | | | | Clients bucketize model | 5 |
| | | Clients compute shares | 1 | Clients compute shares | 1 |
| Server insecurely computes median | 6 | Servers calls C++ backend to compute median | $\sim 500$ | Servers calls C++ backend to compute median buckets | $\sim 125$ |
| | | | | Server quantizes median buckets | 5 |
| **Total** | **30** | **Total** | **525** | **Total** | **160** |

Table 1: CNNMnist, 8 clients, simulated LAN network

In Table 1, we detail the time in seconds each part of the FL process takes for both the pairwise comparison based median method, and our proposed bucketed median method. The bucketed method had the additional overhead on client and server side of converting to and from a bucketed format, and this takes around 5 seconds per transformation for a CNNMnist model. The time for both methods are dominated by the two server secure computation of median. We see that the pairwise comparison based median FL system increases by a factor of $17.5$, and the bucketed median FL system increases by a factor of $5.5$.

In Figure 12, we show the scalability of our protocol when the number of clients range from 10 to 160 in increments of 30. We detail the number of clients on the x-axis and the server-side computation time (after receiving all client updates) in milliseconds on the y-axis. The server-side computation consists of the following parts: 1. adding client shares to compute a histogram for each dimension, 2. performing MPC computation of the bucketed median for each dimension, 3. converting the chosen median buckets to a global model. Steps 2. and 3. are independent of the number of clients. We test on the MLP model. We see that the time for bucketed median computation stays constant, yet the time taken for steps 1-3 roughly increases when number of clients increase due to the time taken for step 1.

In Figure 13, we show the performance of our bucketed protocol with different parameter values. In our protocol, we determine bucket range as $B = \{p_0, \|w'_g - w_g\|_1 + p_{1,t}\}$, where $p_0$ is chosen during the initial epoch, and $\|w'_g - w_g\|_1 + p_{1,t}$ is chosen in subsequent epochs. In our convergence analysis, we use $p_1$ and prove convergence for this choice. In our experiments, we use $p_t = \frac{p_1}{t}$, where $t$ is the current epoch count. Both choices empirically result in decreasing training loss per epoch, albeit at slightly different rates. We note that in our implementation, we set both $p_0$ and $p_1$ as

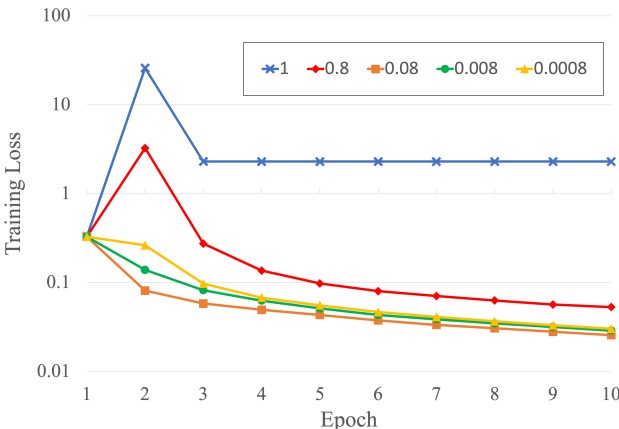

Figure 13: Variable parameter values for bucket range tested on a three client FL system and CNN-Mnist model.

equivalent values, and use the CNNMnist model. For parameter values less than 1, we see that our bucketed mechanism results in a similar rate of training loss convergence. For each model, one can solve for specific $p_0, p_1$ values for the best convergence rate, but as the convergence behaviour does not strictly depend on the parameter values, we do not focus our attention towards this optimization.

