# OpenReview forum: "Scalable Robust Federated Learning with Provable Security Guarantees"
_ICLR.cc/2022/Conference — ICLR 2022 Submitted_

### Official Review · Reviewer_6dep · 2021-10-30

**Correctness:** 2
**Technical Novelty And Significance:** 3
**Empirical Novelty And Significance:** 3
**Recommendation:** 3
**Confidence:** 3

**Main Review:**



The paper is understandable but contains quite a few language mistakes and suboptimal formulations.

The technical details (often in appendix) contain quite a few mistakes, it is unclear whether it is possible to fix them all while sticking to the current results.

For example, the robustness theorem (6.1) looks suspicious to me.  As it is formulated, there is no specification of how the adversarial vectors are generated.  There may exist schemes to generate these adversarial vectors which are compatible with the theorem formulation but cause a deviation in the median much larger than what the theorem promises.



A few details:

* Sec 4.2: In order to send their updates to the servers, clients additively secret-share their updates and sends one share to each server. -> send (plural)

* Algorithm 2, part 2, step 2. (d) : where is b defined ?  Or do you mean j instead of b ?

* In algorithm 1, 'b' is the number of buckets (a different meaning than the symbol 'b' in algorithm 2, it is preferable to avoid the same symbol having different meanings at different places).

* In Section 5, "instead of sending actual updates to the server, each client will send a unit vector that has a 1 in the bucket containing the client’s update, a": it therefore seems that as soon as the actual update has fewer bits than the number of buckets the original non-bucketed version may be cheaper from a communication point of view?

* Section 6.1: "Data individual loss function is denoted" -> an article is missing
* "if ratio of adversarial clients is" -> an article is missing for "ratio"

* We provide similar result -> a similar result

* Equation (3): I would have expected \|Med(\{\nabla {\hat F}_i(w) \}_{i=1}^{n_1} \cup \{v_i\}_{i_1}^{n_2}) - \nabla F(w) \|_\infty = \|Med(\{\nabla {\hat F}_i(w) - \nabla F(w) \}_{i=1}^{n_1} \cup \{v_i - \nabla F(w) \}_{i_1}^{n_2}) \|_\infty, but in Eq (3) in the paper the rightmost term of the righthandside of the equation doesn't have "- \nabla F(w)".

* Equation (4), please define (or remind the definition of) a_i.



**Summary Of The Paper:**


The paper proposes to use the median of model updates instead of the average of model updates in federated learning, and proposes an algorithm to perform the computation in a secret-sharing fashion by two independent servers.


**Summary Of The Review:**


While the topic is interesting and the paper may have a valuable contribution, I don't succeed to verify the proofs of its theorems, and the way in which the theorems are stated lets me believe it is possible they are not correct.

---

> ### Author Response · Authors · 2021-11-23
> **Response to Reviewer 6dep**
>
> We thank the reviewer for the comments they provide on our paper, and reply to them one by one. We also thank the reviewer for catching typos in our paper, which we have corrected in the revision.
>
> **[Generating adversarial vectors]**
>
> Theorem 6.1 works for any adversarial vectors, and thus we don’t specify how the adversarial vectors are generated. It is not clear to the authors and it is not specified by the reviewer what may be the example that breaks the theorem.
>
> **[Communication cost in bucketing method]**
>
> In our experiments, communication is not a bottleneck. It is the 2PC on servers to compute dimension-wise median which is the bottleneck. The time it takes for communication between server and client in a LAN is shadowed by the server to server computation time. For example, the server to client communication of a CNNMnist model with 1,663,370 dimensions takes ~20 megabytes. The client to server communication of the share of the bucketed model is ~106 megabytes. In a LAN network with bandwidth of 377 megabytes/second, we see that the communication cost between server and client is minimal with respect to the two-server median computation cost.
>
> **[Typo in equation 3]**
> Thanks for pointing this out. It is a typo and our theories are correct. We have updated the paper with revision of this typo.

---

### Official Review · Reviewer_SNQM · 2021-11-01

**Correctness:** 3
**Technical Novelty And Significance:** 3
**Empirical Novelty And Significance:** 3
**Recommendation:** 5
**Confidence:** 4

**Main Review:**

Strengths

1. The topic is interesting. Achieving privacy and robustness is an important goal for federated learning.

2. An approximate median aggregation rule is proposed. The proposed approximate median is crypto-friendly.

3. Some theoretical analysis about the approximate median is proposed.

Weaknesses

1. The proposed crypto method assumes two non-colluding servers. This assumption may be unrealistic.

2. The compared baseline seems too simple. Basically pairwise comparison is adopted as a baseline for implementing the standard median. How about median of median? We can divide the clients into groups, calculate median in each group, and then take median of median. This is another approximate method and is more efficient.

3. The evaluated Byzantine failures are too simple. Basically,  bit flips failure, label flip failure, and Gaussian noise failure are considered. I suggest the authors to consider more advanced attacks, e.g., the following:

a. A Little Is Enough: Circumventing Defenses For Distributed Learning. In NeurIPS, 2019.

b. Local Model Poisoning Attacks to Byzantine-Robust Federated Learning. In Usenix Security Symposium, 2020.

If my comments, especially 2 and 3 are addressed, I'm happy to increase my rating score.  Overall, I think this is a promising direction. We can pick a robust federated learning method and then use cryptography to make it privacy preserving. However, we need to pick a really robust federated learning method. The paper picks Median, which shows some robustness but is not really robust. I think picking a method with provable robustness guarantee would make the paper stronger, e.g., the following:

c. Provably Secure Federated Learning against Malicious Clients. In AAAI, 2021.

d. CRFL: Certifiably Robust Federated Learning against Backdoor Attacks. In ICML, 2021.



**Summary Of The Paper:**

This paper aims to build privacy-preserving and robust federated learning systems. The proposed method includes two key ideas. The first idea is to propose an approximate, crypto-friendly median aggregation rule, which aims to achieve robustness. The second idea is to use cryto methods to implement this aggregation rule. Experiments were conducted to show that the proposed method is more efficient than cryto-based standard median implementation.

**Summary Of The Review:**

The paper studies privacy-preserving and robust federated learning. I think the paper would be stronger if my comments are addressed.

---

> ### Author Response · Authors · 2021-11-23
> **Response to Reviewer SNQM**
>
> We thank the reviewer for the comments they provide on our paper, and reply to them one by one.
>
> **[Weakness 1. Non-colluding servers]**
>
> Two non-colluding servers is a common assumption for distributed computing using MPC [1, 2], especially when it comes to methods that focus on scalability, since then computation will be in 2PC. There are many papers in the distributed learning field which utilize this assumption to attain efficient results [3, 4, 5, 6]. Furthermore, data privacy legal regulations such as GDPR [7] show that two non-colluding servers are a realistic assumption. Additionally, there are other incentives for different entities to not collude: keeping customer data within the institution due to business competitors, reputation of institutions if unauthorized collaboration is made public to their users, etc.
>
> [1] Z. Wei et al. “Efficient Privacy Preserving Cross-Datasets Collaborative Outlier Detection.” CSS 2019.
>
> [2] Alexandru, et al. “Cloud-based MPC with Encrypted Data.” IEEE CDC 2018.
>
> [3] Y. Khazbak et al. "MLGuard: Mitigating Poisoning Attacks in Privacy Preserving Distributed Collaborative Learning." IEEE ICCCN, 2020.
>
> [4] H. Fereidooni, et al. "SAFELearn: secure aggregation for private federated learning." 2021 IEEE Security and Privacy Workshops (SPW).
>
> [5] L. He, et al. “Secure Byzantine-Robust Machine Learning.”
>
> [6] P. Mohassel et al. “SecureML: A System for Scalable Privacy-Preserving Machine Learning.” IEEE Symposium on Security and Privacy 2017.
>
> [7] “General Data Protection Regulation.” 2018, https://eur-lex.europa.eu/eli/reg/2016/679/oj.
>
> **[Weakness 2. Median of median baseline]**
>
> Thank you for the suggestion. We consider pairwise comparison based median as a baseline because that is the most efficient way we can think of to calculate the exact median. Our bucketing based approximate median is more efficient than a median of median based method, when considering number of comparisons as a complexity metric.
>
> We denote n as the number of parties, m as the size of each subgroup to compute the first median on, and b as the number of buckets used in our bucketing scheme. For our bucketing scheme, the number of comparisons needed per dimension is b, empirically we used b=8. In the median of medians scheme, where the first and second level medians are calculate using pairwise comparisons,  the number of comparisons needed per dimension is $\frac{n}{m} * {m \choose 2} + {\frac{n}{m} \choose 2}$. When setting number of parties, n, to be a reasonable value such as 100, we see for the inequality $\frac{n}{m} * {m \choose 2} +{ \frac{n}{m} \choose 2} < 8$ to hold, there is no positive value of m such that this inequality holds.
>
> We agree that our baseline could be more efficient. However, we note depending on the chosen size of the subgroups, median of medians guarantees robustness against a differing number of faulty parties;  For example, choosing subgroup size of 5 leads to a median of median value that is at least larger than 30% and at most larger than 70% of the elements - which is less than the 50% robustness of median. So optimizing the subgroup size of the median of median method for efficiency purposes may lead to robustness against a smaller number of faulty parties than our bucketed median method.
>
> We additionally believe that computing the median of medians as an efficient approximation of median does not exist in prior robustness literature.
>
> **[Weakness 3. Evaluated Byzantine failures are too simple]**
>
> Thank you for the constructive feedback on our work. We also believe that efficient, robust, and private distributed learning is a promising and new direction. There are few works that emphasize efficiency while still providing privacy and some level of robustness. As such, the main goal of our work is to improve on the efficiency of private computation of  some robust aggregation rule (which we believe our work contributes greatly to in our server-side median calculation whose complexity stays constant with relation to number of clients). Our work provides robustness against realistic faults in the model training process that may be caused by benign device or coding errors. Improving robustness is not a main goal of our work, but we are interested in future work to improve the efficiency of private computation of more robust rules that are effective against backdoor attacks.

---

### Official Review · Reviewer_moZd · 2021-11-02

**Correctness:** 4
**Technical Novelty And Significance:** 3
**Empirical Novelty And Significance:** 3
**Recommendation:** 5
**Confidence:** 3

**Main Review:**

The paper considers the two-server federated learning setting and proposes an interesting and simple approach for tolerating malicious users. The protocol is efficient as its complexity grows only linearly in the number of the users, whereas it grows quadratically in prior works.  The approximation is based on bucketing, which divides each dimension of the model into buckets and sends a unit vector that has a 1 in the bucket containing the client’s update and the median is approximated as the middle value of the range of this bucket. My main concern is that the paper does not consider some closely related works. Hence, it is not clear if the proposed approach is better. Below, I provide my detailed comments.

1- It is not clear for me how the proposed approach of this paper compares to other related works that are not mentioned as [1, 2]. The authors should make sure to include all relevant works along with comparisons that show the effectiveness of the proposed approach compared to these works.

2- Typo in Page 2: "showe" -> show

3- Page 4: I find the static corruption model in Section 4.1 to be unrealistic as it assumes that the adversary chooses the parties to corrupt at the beginning of the protocol and cannot change them. In general, the corrupted parties may change dynamically while the protocol is running.

4- Page 6: The convergence analysis only considers the IID case, how about the non-IID case?

5-  The following sentence should be illustrated more perhaps by providing numerical examples.
"Note that the previous analysis on dimension-wise median in the FL setting (Yin et al., 2018) requires α to be smaller than an easily negative value"

6- The number of users considered in the experiments is too small to claim scalability. More experiments with large number of users need to be performed to show the effectiveness of the proposed approach in large-scale systems.

7- Fig. 5 shows large gap between the proposed approach and the median approach, why this is the case?

References

[1] Y. Khazbak et al. "MLGuard: Mitigating Poisoning Attacks in Privacy Preserving Distributed Collaborative Learning." , IEEE ICCCN, 2020.

[2] H. Fereidooni, et al. "SAFELearn: secure aggregation for private federated learning." 2021 IEEE Security and Privacy Workshops (SPW).

**Summary Of The Paper:**

This paper proposes a fast, secure, private and scalable approximate median aggregation approach for federated learning with two semi-honest non-colluding servers. The convergence analysis of the proposed approach for the IID case has been also provided under certain assumptions, which shows that the proposed approach converges with high probability if less than 1/2 of the clients are malicious. Finally, the experiments show that the proposed approach achieves similar robustness performance compared to the exact median based approach.

**Summary Of The Review:**

The paper proposes an interesting approach approach for tolerating malicious users in FL, but it does not consider some prior works. Hence, I recommend revising the paper to include all relevant works and adding the comparisons.

-- Post Rebuttal  --

I still do not see how the proposed approach in this paper compares to the related works I mentioned. While [1] does not provide server side running times, you should still evaluate their protocol and compare with your work. The responses to the non-IID issue and the scalability issue that I raised are not satisfying neither. Hence, I am keeping my score.

---

> ### Author Response · Authors · 2021-11-23
> **Response to Reviewer moZd**
>
> We thank the reviewer for the comments they provide on our paper, and reply to them one by one.
>
> **[1. Comparison with related works]**
>
> We thank the reviewer for pointing out these two works that are related to the problem of private and efficient FL. Below, we note the differences we believe our work intends to solve compared to the problems [1,2] intend to solve.
> [1] proposes an efficient (in terms of MPC operations) privacy preserving robust FL protocol that uses a ranking system based on cosine similarity. One difference is that they use the cosine similarity (not approximate median) method to protect against data poisoning attacks from adversarial clients.
>
> The results of [1] focus on the robustness of their proposed protocol. The authors of [1] only provide times for client side computation and do not provide server side running times for a round of aggregation computation. Furthermore, they focus on the setting of low powered mobile clients, meaning client dropout may occur, which differs from our multi-silo setting where we assume dropouts do not happen during execution of the protocol.
>
> [2] consider the problem of communication required for one round of Federated Learning which uses secure averaging as a private aggregation rule. Their main contribution is reducing the number of rounds of communication of one epoch of FL to 2 when there are client dropouts. They do not focus on efficient cryptographic protocol design for private median computation.
>
> [1] Y. Khazbak et al. "MLGuard: Mitigating Poisoning Attacks in Privacy Preserving Distributed Collaborative Learning." , IEEE ICCCN, 2020.
>
> [2] H. Fereidooni, et al. "SAFELearn: secure aggregation for private federated learning." 2021 IEEE Security and Privacy Workshops (SPW).
>
> **[3. Static Corruption ]**
>
> In our static corruption model, the “static-ness” corresponds to this number of corrupted clients, not the clients themselves. Thus the scenario of corrupted clients changing dynamically between rounds is still captured by our “static” corruption model, as long as the number of corrupted clients remains below the constant threshold at each round. However, we restrict the adversary to only choosing one of the servers to corrupt through the entire protocol; efficient MPC in the adaptive corruption model is still an open problem [1].
>
> [1] A. Chopard et al. “On Communication-Efficient Asynchronous MPC with Adaptive Security.” IACR TCC 2021
>
> **[4. Non-IID convergence]**
>
> The IID assumption is a standard assumption for the convergence analysis of federated learning, especially when the focus is robustness. (e.g. [1, 2]). Convergence analysis for federated learning with non-IID data is an active research direction even for the standard FedAvg (e.g. [3, 4]).
>
> [1] Dong Yin, et al. Byzantine-robust distributed learning: Towards optimal statistical rates. ICML 2018.
>
> [2] Cong Xie, et al. Zeno: Distributed stochastic gradient descent with suspicion-based fault-tolerance. ICML 2019.
>
> [3] Li, Xiang, et al. On the Convergence of FedAvg on Non-IID Data. ICLR 2020.
>
> [4] Margalit Glasgow et al. Sharp Bounds for Federated Averaging (Local SGD) and Continuous Perspective. 2021.
>
> **[5. Comparison with Yin et al., 2018]**
>
> Thanks for the suggestion. As shown in the equation 2 of (Yin et al. 2018), they assume the ratio of adversary clients $\alpha$ to be smaller than $\frac{1}{2}-C$ where the $C$ is greater than $\frac{1}{2}$ unless the model is extremely smooth. For example, let’s suppose the dimension of the model to be $1000$ and the model parameters are within a Euclidean ball of radius 1. Also, suppose there are $10$ clients where each has $1000$ data. Then, the $C>\frac{1}{2}$ if the model smoothness is greater than $10^{-6}$. The smoothness is required to be even smaller for models bigger than this example.
>
> **[6. Scalability Results]**
>
> Thank you for the suggestion. The scenario we work in is multi-silo, which we assume to be implemented with a relatively small number of clients (tens to hundreds). To this end, we provide results of our bucketed median scheme on a larger number of clients in Appendix F of our revision. We test with 10 - 160 clients in increments of 30, and we see that the server-side computation for dimension-wise bucketed median stays roughly constant.
>
> **[7.  Gap between bucketing and median in Fig. 5]**
>
> This is because the x-axis is the epoch number on the graph. Strictly looking at a per-epoch case, the approximate median calculated by our bucketing scheme obtains lower accuracy than a median method. However, each epoch takes less time for our method.

---

### Official Review · Reviewer_mthn · 2021-11-13

**Correctness:** 3
**Technical Novelty And Significance:** 2
**Empirical Novelty And Significance:** 1
**Recommendation:** 5
**Confidence:** 3

**Main Review:**

The problem of achieving security as well as Byzantine robustness in federated learning is challenging, and there are only a few papers on this topic. In this sense, the paper is timely and relevant. However, the threat model considered in the paper is non-standard (and also a bit confusing). Furthermore, the novelty seems to be limited as the paper mainly combines two ideas from the literature: bucketing (from [Corrigan-Gibbs and Boneh 2017]) and secure median ([Tueno et al. 2019]). The detailed comments are as follows:

Major comments:

1. In the majority of papers that consider robust federated/distributed learning, it is assumed that Byzantine/adversarial clients can arbitrarily deviate from the protocol. The paper mentions in the abstract about adversarial updates, but also assumes that all clients honestly follow the protocol. This is somewhat non-standard threat model. Is it assumed that errors occur due to external factors such as hardware faults etc? If this is the case, then errors may occur while computing secret shares of the model as well. It is assumed that secret shares are computed perfectly, which does not seem practical. Also, first mentioning adversarial updates and then mentioning semi-honest clients can be a bit confusing. It would be important to add a discussion on this model, and explicitly mention how this differs from Byzantine or malicious setup considered in prior works.

2. The title of the paper says scalable robust federated learning, however experiments are performed with just 3 clients. This seems to be a bit too limited. Since MPC primitives will mainly be used between the two servers and clients simply secret share their updates, it is not clear why not to scale the protocol to a larger number of clients in the experiments. At the very least, can the authors comment on how would the bucketed median perform (without security constraints) for a larger number of clients, when a constant fraction of clients can be faulty?

3. Assumption 2 seems to require that each client has a very large dataset, and computes gradient descent on the entire dataset. Can the authors elaborate on this? This seems to limit the practicality. How the assumptions in the paper differ from typical assumptions, say those in [Yin et al. 2018]?

Taking this point further, the paper says (before Theorem 6.1) that [Yin et al. 2018] requires \alpha to be "smaller than an easily negative value". Can the authors give more details on what this means? Further, the authors say that they prove the robustness in a new way.  It will be helpful to elaborate what new techniques are used, and how they differ from [Yin et al. 2018].

4. In Algorithm 1, it is not at all clear how F^l_{Comp}(<\sum_{i}e_i>_j, n/2) can compute the median on buckets. Specifically, why does  the comparison algorithm take sum of shares as input? In general, it is a bit unclear how the frequency distribution of client updates is computed, or even how it is defined. It is important to give sufficient details here.

5. In Experimental Results, it would be important to give more details about the models considered. For instance, in the CNN for CIFAR-10, how many convolutional layers, what pooling etc.

Other comments:

1. The authors mention that best known approaches for median aggregation with MPC are slow, having quadratic computation cost (in terms of the number of clients). There are no citations given. Can the authors provide some references to support this claim?

2. The paper cites [Konecny et a. 2017] for MPC in federated learning at two instances in the first paragraph. This is incorrect as [Konecny et a. 2017] deals with reducing communication costs, and does not consider MPC.

3. Can the proposed protocol handle client dropouts? It would be helpful to mention this explicitly.

4. Secure comparison is a significantly complex computation. The complexity of the proposed algorithm is characterized in terms of number of calls to secure comparison. It would be helpful to elaborate on the complexity of secure comparison to give a fair idea to readers (who are less aware of MPC primitives).

5. The paper claims in the title that the proposed algorithm has provable security guarantees, but spends only a paragraph in the main paper about the security analysis. It would be helpful to add more details, as security considerations are important for the problem setup.

6. For label flipping, computing mean seems to perform better than computing median. It would be helpful to add a comment on why this is the case.

**Summary Of The Paper:**

The paper considers federated learning with two non-colluding honest-but-curious servers and honest-but-curious clients, where a subset of clients can be faulty. It proposes a protocol to securely compute approximate median. The main idea is to use secure median algorithm from [Tueno et al. 2019] along with a bucketing trick proposed in [Corrigan-Gibbs and Boneh 2017] to reduce complexity. The paper gives convergence analysis for bucketed median under some strict assumptions.

**Summary Of The Review:**

The novelty seems to be fairly limited as the paper mainly combines two known ideas: bucketing and secure median. The threat model is somewhat non-standard, and experiments are conducted on only 3 clients.

---

> ### Author Response · Authors · 2021-11-23
> **Response to Reviewer mthn**
>
> We thank the reviewer for the comments they provide on our paper, and reply to them one by one.
>
> **[Major 1] Clarification on security model**
>
> To clarify our security model, we can split our protocol into two parts: (1) model training and (2) private median computation. We assume that faults are only introduced in the model training part.
> We do not assume that our protocol is robust against Byzantine parties. [1] show that dimension-wise median is not robust to byzantine parties. However, median is still robust to faults that arise from communication and coding errors, as highlighted in prior work (e.g., Yin et al. 2018), as we demonstrate in our experiments section.
>
>
> [1] C. Xie et al. “Fall of empires: Breaking Byzantine-tolerant SGD by inner product manipulation.” Uncertainty in Artificial Intelligence, 2020.
>
> **[Major 2] Testing with more clients**
>
> We thank the reviewer for the feedback. In Appendix F of our revision, we provide the results of testing with 10-160 clients in increments of 30. We note that the time taken for the private computation between the two servers of pairwise median stays roughly constant.
>
> In our convergence proof, we show that the faulty clients only have limited impact on the bucketed median when the number of faulty clients is strictly less than half.
>
> **[Major 3] Elaboration on Assumption 2**
>
> The assumption 2 (weak law of large number) is a rather standard assumption in statistics, i.e., it only requires “sample mean converges to the true mean”. We do not assume the speed of convergence as it is more general. It only requires a small dataset if the convergence speed is fast (e.g., the variance is small), but making an assumption on the variance is stronger than the weak law of large number. We only use the gradient on the entire dataset in the convergence analysis, and stochastic gradient descent can be applied in practice.
>
> **On relation to [Yin et al. 2018]**
>
> The related assumptions in [Yin et al. 2018] are pointwise bounded variance and skewness, while ours is the uniformly weak law of large number. The two sets of assumptions are not directly comparable: our assumption is weaker in the sense that we do not assume bounded variance and skewness, and our assumption is stronger in the sense that it is made in an uniform setting.
>
> However, their assumptions are not sufficient, thus their requirements on $\alpha$ are strong. As shown in the equation 2 of (Yin et al. 2018), they assume the ratio of adversary clients $\alpha$ to be smaller than $\frac{1}{2}-C$ where the $C$ is greater than $\frac{1}{2}$ unless the model is extremely smooth. For example, let’s suppose the dimension of the model to be $1000$ and the model parameters are within a Euclidean ball of radius 1. Also, suppose there are $10$ clients where each has $1000$ data. Then, the $C>\frac{1}{2}$ (i.e., $\alpha<0$) if the model smoothness is greater than $10^{-6}$. The smoothness is required to be even smaller for models bigger than this example.
>
> Our choice of the assumption, not stronger than theirs, helps us avoid the vacuous requirements on $\alpha$, i.e., our results hold for any $\alpha<\frac{1}{2}$.
>
>
> **[Major 4] Clarification on comparison ideal functionality**
>
> Summing up the e_i from each client results in a histogram per dimension of the model. The ideal functionality takes this vector which contains histograms for each dimension, and checks for each dimension which bucket contains the n/2 th value. We kindly point the reviewer toward Section 5 for a succinct description of the protocol and Appendices A and C for a more detailed explanation of the protocol.
>
> **[Major 5] Model details**
>
> We have added model details to the revision of the paper in the experiments section.
>
>
> **[Other 1] Best known MPC median**
>
> In Section 4.2, we provide [1] as a work that focused on efficiently calculating median in MPC.
>
> [1] A. Tueno et al. “Secure computation of the kth-ranked element in a star network”, International Conference on Financial Cryptography and Data Security, 2019.
>
> **[Other 2] Incorrect citation**
>
> We thank the reviewer for pointing this out, and we have made the appropriate changes.
>
> **[Other 3] Client dropout**
>
> We are working in a multi-silo setting (hospitals, large institutions) and not in a mobile client setting and thus do not consider client dropouts.
>
>
> **[Other 4] Secure comparison**
>
> We agree with the reviewer and have included a brief discussion in Appendix C.1 of our revised paper on elaborating on the inefficiency of secure comparisons.
>
>
> **[Other 5] Security proof**
>
> We provide a formal simulation-based security proof in Appendix B.
>
>
> **[Other 6] Label flipping mean vs. median**
>
> We note that [1] show similar results in the convergence rate of mean and median methods with label-flipping faults.
>
> [1] C. Xie et al. “Zeno: Distributed Stochastic Gradient Descent with Suspicion-based Fault-tolerance” Proceedings of the 36th International Conference on Machine Learning, 2019.

---

> > ### Comment · Reviewer_mthn · 2021-11-30
> > **Thanks for responding!**
> >
> > Thanks to the authors for their responses. The responses address some of my concerns. I have two further suggestions:
> > * I would suggest the authors to explicitly mention the threat model, along with the justification.
> > * The authors include a discussion on the computation costs of secure comparison. It would be great to also include a discussion on communication costs. Even though the protocols in [Damgard et al., 2006] and [Nishide and Ohta, 2007] take constant-rounds of communication, the actual number of rounds is pretty large.
> >
> > Overall, I still think the novelty is somewhat limited because the paper mainly combines two known ideas. Further, the experiments on larger number of clients in Appendix F only measure computation time at servers. Machine learning experiments are still restricted to a very small number of clients. Considering this, I am raising my score to 5.

---

> > > ### Author Response · Authors · 2021-12-03
> > > **Thank you for the suggestions**
> > >
> > > We would like to thank the reviewer for their timely response and reconsideration of our paper. Below are our responses to the suggestions the reviewer brings up.
> > >
> > > **[Threat Model and Justification]**
> > >
> > > Threat Model: We achieve security according to Appendix B.1 Def. 1 against static, semi-honest corruption of one server and upto all-but-one of the clients. We achieve robustness against faults in upto half the clients in every round, and the set of faulty clients can change between rounds. We note that here we crucially differentiate between client corruptions in the semi-honest setting and client faults. The former must be static, and can impact upto all-but-one clients, while the latter can change dynamically across rounds.
> > >
> > > Justification: Our security model assumes only semi-honest corruptions; while weaker than security against malicious corruptions, this is a realistic model capturing an adversary that wants to secretly eavesdrop but will otherwise follow protocol instructions.
> > > We only allow faults in the model training phase of our protocol in order to show its robustness guarantees.
> > >
> > > **[Secure comparison communication costs]**
> > >
> > > We will edit our current version of the paper to also include a discussion on the communication costs of secure computation in the Appendix. We will discuss that even though rounds are constant, this constant could be high, and the actual communication cost could still be quite large.
> > >
> > > **[Number of clients]**
> > >
> > > We agree that the number of clients we test with may seem small for the setting of distributed learning when clients are mobile devices. However, we note that the setting we are working with is the multi-silo setting, which assumes a smaller number of clients similar to the number we have tested with.
> > > Works in robustness are more concerned with the ratio of adversarial clients compared to the number of clients. [1,2] perform experiments with 10 clients.
> > >
> > > [1] A. Reisizadeh, et al. “Robust Federated Learning: The Case of Affine Distribution Shifts.” NeurIPS 2020.
> > >
> > > [2] Y. Deng, et al. “Distributionally Robust Federated Averaging.” NeurIPS 2020.

---

### Decision · Program_Chairs · 2022-01-20

**Decision:**

Reject

**Comment:**

There were concerns that the paper has a fairly limited novelty, being based on the combination of two known ideas: bucketing and 2-party secure median for distributed learning. Also, the scale of experiments is quite limited. Other issues include the lack of comparison to relevant related work, some doubts on correctness, and issues with independence and scalability that weren't fully resolved. Overall the reviewers felt that the paper shoud not be accepted in its current form.